# AbBiBench: A Benchmark for Antibody Binding Affinity Maturation and Design

## Abstract

We introduce **AbBiBench** (**A**nti**b**ody **Bi**nding **Bench**marking), a benchmarking framework for antibody binding affinity maturation and design. Unlike previous strategies that evaluate antibodies in isolation, typically by comparing them to natural sequences with metrics such as amino acid recovery rate or structural RMSD, AbBiBench instead treats the antibody–antigen (Ab–Ag) complex as the fundamental unit. It evaluates an antibody design's binding potential by measuring how well a protein model scores the full Ab–Ag complex. We first curate, standardize, and share more than 186,580 experimental measurements of antibody mutants across 13 antibodies and 9 antigens—including influenza, lysozyme, HER2, VEGF, integrin, Ang2, and SARS-CoV-2—covering both heavy-chain and light-chain mutations. Using these datasets, we systematically compare 15 protein models including masked language models, autoregressive language models, inverse folding models, diffusion-based generative models, and geometric graph models by comparing the correlation between model likelihood and experimental affinity values. Additionally, to demonstrate AbBiBench's generative utility, we apply it to antibody F045-092 in order to introduce binding to influenza H1N1. We sample new antibody variants with the top-performing models, rank them by the structural integrity and biophysical properties of the Ab–Ag complex, and assess them with in vitro ELISA binding assays. Our findings show that structure-conditioned inverse folding models outperform others in both affinity correlation and generation tasks. Overall, AbBiBench provides a unified, biologically grounded evaluation framework to facilitate the development of more effective, function-aware antibody design models.

## 1 Introduction

Antibodies are critical components of the adaptive immune system, functioning primarily by recognizing and binding specifically to antigens such as pathogens or aberrant cells. This specific recognition is facilitated through complementary regions: the antibody's paratope and the antigen's epitope. Improving antibody–antigen affinity boosts therapeutic potency and is critical for drug discovery. Developing a therapeutic monoclonal antibody depends on multiple factors, including expression, stability, immunogenicity, aggregation, and binding affinity Chungyoun et al. (2024); Jain et al. (2017). Among these, binding affinity is the most critical determinant of therapeutic efficacy, as it directly influences antibody potency. Therefore, increasing binding affinity between antibody and antigen has been a crucial process in therapeutic antibody development.

Traditional antibody discovery methods, such as phage display technology Marks et al. (1992); McCafferty et al. (1990); Smith (1985) and animal immunization Green et al. (1994); Köhler & Milstein (1975), employ iterative cycles of mutation and selection to progressively improve binding affinity. These methods have significantly advanced therapeutic antibody discovery but face challenges due to the initially limited diversity of libraries in phage display or the narrow naive B cell repertoire available in animal immunization models, restricting the comprehensive exploration of potential high-affinity antibody variants (Fig. 1a). Machine learning-based antibody design approaches complement these experimental techniques by efficiently navigating the vast search space (Fig. 1b) and proposing high-affinity antibody variants that may not be readily accessible through experimental approaches alone.

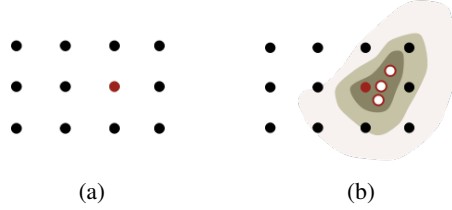

Figure 1: Antibody design space. (a) Traditional *in vitro* screening explores a limited antibody library. (b) Protein machine learning models explore broader mutational space. ●: tested antibodies, ●: lead antibody, ○: model-generated variants.

Recent advances in machine learning, especially with protein language models and structure-based generative models Dauparas et al. (2022); Hayes et al. (2025); Høie et al. (2025); Hsu et al. (2022); Kong et al. (2022; 2023); Li et al. (2024); Luo et al. (2022); Malherbe & Ucar (2024); Martinkus et al. (2023); Ruffolo et al. (2021); Shanker et al. (2024); Su et al. (2023); Watson et al. (2023); Wu & Li (2023), have shown promise in antibody design. However, common evaluation metrics like amino acid recovery rates or structural RMSD to natural antibodies do not adequately capture biological relevance in antibodies. In general protein design, comparing to the closest natural variant is a reasonable validation strategy because protein mutations are driven by strong evolutionary pressure. By contrast, antibody generation involves stochastic recombination and hypermutation, yielding extreme diversity. Even antibodies to the same antigen often show little sequence similarity unless clonally related. As such, evaluating designed antibodies by how closely they resemble naturally occurring ones overlooks the fundamental biology of antibody generation. This calls for new evaluation criteria that better reflect the functional goals of antibody engineering, rather than assumptions borrowed from general protein design.

From a structural biology perspective, binding affinity is determined not just by the antibody sequence, but by the quality of the interface it forms with the antigen. High-affinity binding typically arises from antibody-antigen (Ab-Ag) complexes that exhibit structural integrity Shanker et al. (2024) – meaning they are stable, well-packed, and maintain favorable conformations with minimal strain. Structural integrity ensures optimal shape and chemical complementarity at the binding interface. Antibodies that form such stable complexes with their targets resemble naturally occurring Ab-Ag complexes such as those collected in structural databases like SAbDab Dunbar et al. (2014). Recent machine learning models Dauparas et al. (2022); Hsu et al. (2022); Li et al. (2024); Luo et al. (2022); Malherbe & Ucar (2024); Su et al. (2023); Wu & Li (2023) have shown success in learning Ab-Ag complex sequence-structure patterns, enabling us to gauge whether a designed Ab-Ag complex lies within the high-probability manifold of structurally stable, high-affinity complexes. Therefore, incorporating the antigen into evaluation provides a more biologically grounded and functionally relevant assessment of antibody design.

To address the limitations discussed above, we introduce AbBiBench (**A**nti**b**ody **Bi**nding **Bench**marking), a biologically relevant benchmarking framework specifically designed for improving antibody binding affinity. Rather than assessing antibodies in isolation Chungyoun et al. (2024), we consider the Ab-Ag complex as a unit for evaluation. We curated standardized data from publicly available experimental binding affinity studies, compiling 186,580 mutated antibodies across nine antigen targets to evaluate protein models for binding affinity optimization.[1] We also devised and publicly shared an efficient pipeline to rank newly designed antibodies based on complex structural integrity and biophysical properties.[2] AbBiBench is curated to avoid data leakage: although wild-type antibodies or antigens may appear in public datasets, no training corpus contains the mutant antibody–antigen complexes it evaluates. By providing a rigorous, biologically grounded benchmark for antibody design, AbBiBench will accelerate methods that lead to clinically and diagnostically impactful discoveries.

## 2 RELATED WORK

### 2.1 MEASURING BINDING AFFINITY BETWEEN ANTIBODY AND ANTIGEN

Three main approaches for measuring binding affinity exist: experimental Abdiche et al. (2008); Jönsson et al. (1991); Livingstone (1996), biophysics-based Buß et al. (2018); Chi et al. (2024); Weitzner et al. (2017), and data-driven (machine learning) methods Dauparas et al. (2022); Evans

---

[1]https://huggingface.co/datasets/AbBibench/Antibody_Binding_Benchmark_Dataset
[2]https://github.com/MSBMI-SAFE/AbBiBench

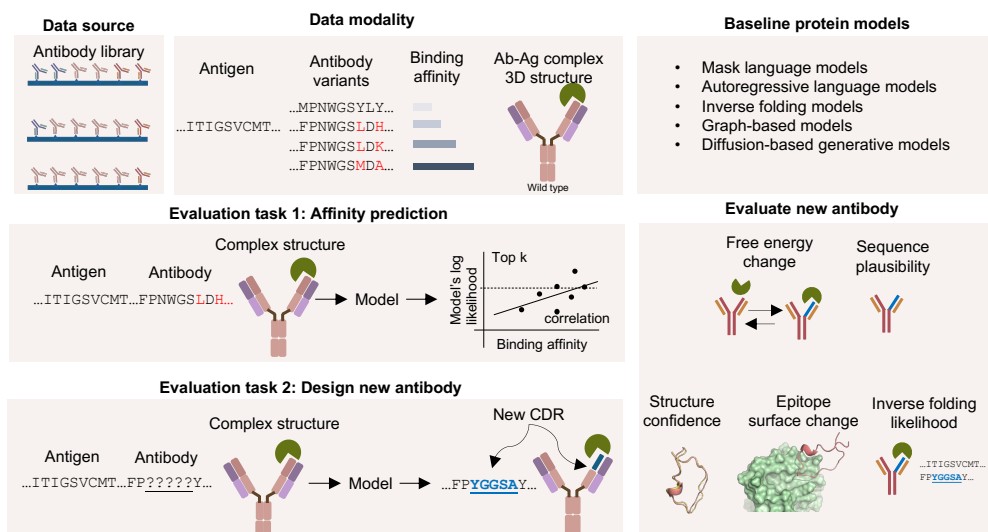

Figure 2: Overview of AbBiBench benchmarks. Antibody variants with experimentally determined affinity values are curated. Data modalities include amino acid sequences, wild-type antibody-antigen complexes, and affinity scores. A diverse set of baseline models includes general protein language models and specialized antibody models. All models are evaluated on two tasks: affinity prediction and antibody redesign. Five computational metrics assess newly designed antibodies from sequence plausibility, structural integrity, and binding affinity perspectives.

et al. (2021); Jumper et al. (2021); Lin et al. (2023). The most direct method to measure binding affinity is through the dissociation constant (Kd), which experimentally quantifies how tightly an antibody binds to its antigen. They are costly and low-throughput, thus computational models aim to approximate the binding affinity to enable large-scale screening. Biophysics-based computational models estimate Ab-Ag binding affinity by calculating interaction energies and free energy change, such as FoldX and $\Delta\Delta G$. It provides mechanistic insights into binding affinity. Though not direct affinity estimator, structural integrity (or plausibility) of Ab-Ag complex have been used as a proxy for binding affinity, measured via structure-based confidence metrics (interface pLDDT, iPTM) Evans et al. (2021); Jumper et al. (2021) or ML-based perplexity (e.g., perplexity from AntiBERTy, ProteinMPNN, or ESM-IF) Dauparas et al. (2022); Lin et al. (2023); Ruffolo et al. (2021). Despite the efforts to measure binding affinity computationally, the reliability and accuracy of these metrics remain an ongoing challenge Johnson et al. (2025).

## 2.2 PRIOR BENCHMARK STUDIES EVALUATING BINDING AFFINITY VIA MODEL LIKELIHOOD

Alternatively, some studies retrospectively evaluate machine learning models using historical experimental protein fitness data Chungyoun et al. (2024); Ucar et al. (2024); Wang et al. (2023). These works assume that higher model likelihoods correlate with higher protein fitness, suggesting the model can generate functional sequences. ProteinGym Notin et al. (2023) exemplifies this approach by benchmarking models against experimental data on enzymatic activity, binding, expression, and stability. It measures correlation between model perplexity and fitness scores. However, ProteinGym does not include antibodies. FLAb Chungyoun et al. (2024) addresses this gap by compiling experimental measurements specifically for antibodies, including binding affinity, specificity (e.g., polyreactivity), immunogenicity, and developability metrics. While comprehensive, FLAb does not consider the antigen when evaluating binding affinity. This is a critical limitation, as antibody-antigen interactions are highly specific, and accurate affinity prediction requires modeling the Ab-Ag complex. BindingGYM Lu et al. (2024) is a a related data curation effort for protein-protein interactions. While BindingGYM is a valuable resource, our work differs in scope and construction. We are specifically interested in interactions that involve an antibody's CDR loops and an antigen and for this reason, only 6 Ab-Ag pairs out of BindgingGYM's 30 overlap with those we have curated here.

| ID | Antibody | Antigen | Variants | Binding score | Study |
|---|---|---|---|---|---|
| 4fqi_h1 | CR9114 | Influenza A/ New Caledonia/20/99 (H1N1) | HC: 65,094 | $-\log K_d$ | Phillips et al. (2021) |
| 4fqi_h3 | CR9114 | Influenza A/ Wisconsin/67/2005 (H3N2) | HC: 65,535 | $-\log K_d$ | Phillips et al. (2021) |
| 3gbn_h1 | CR6261 | Influenza A/ New Caledonia/20/99 (H1N1) | HC: 1,887 | $-\log K_d$ | Phillips et al. (2021) |
| 3gbn_h9 | CR6261 | Influenza A/ Hong Kong/1073/1999 (H9N2) | HC: 1,842 | $-\log K_d$ | Phillips et al. (2021) |
| aayl49 | AAYL49 | Spike HR2 | HC: 4,312 | $-\log K_d$ | Engelhart et al. (2022) |
| aayl49_ML | AAYL49_ML | Spike HR2 | HC: 8,953 | $-\log K_d$ | Li et al. (2023) |
| aayl50 | AAYL50 | Spike HR2 | LC: 11,473 | $-\log K_d$ | Engelhart et al. (2022) |
| aayl51 | AAYL51 | Spike HR2 | HC: 4,320 | $-\log K_d$ | Engelhart et al. (2022) |
| aayl52 | AAYL52 | Spike HR2 | LC: 13,324 | $-\log K_d$ | Engelhart et al. (2022) |
| 2fjg | G6.31 | VEGF | HC: 2,223 LC: 2,014 | $log$ enrichment | Koenig et al. (2017) |
| 1mlc | D44.1 | Hen-egg-white lysozyme | HC: 1,229 LC: 865 | $log$ enrichment | Warszawski et al. (2020) |
| 1n8z | trastuzumab | HER2 | HC: 419 | $-\log K_d$ | Shanehsazzadeh et al. (2023) |
| 1mhp | AQC2 | Integrin-$\alpha$-1 | HC: 37 LC: 25 Both: 5 | $-\log K_d$ | Clark et al. (2006) |
| 4d5_her2 | 4D5 | HER2 | Both: 2,080 | $log$ enrichment | Minot & Reddy (2024) |
| 5a12_ang2 | 5A12 | Ang2 | HC: 796 LC: 104 Both: 43 | $log$ enrichment | Minot & Reddy (2024) |

Table 1: Overview of the 15 Ab-Ag binding-affinity assays reported in AbBiBench, showing the number of heavy-chain mutants in each study and the respective binding metric.

# 3 ABBIBENCH: AN ANTIBODY BINDING BENCHMARK

We introduce AbBiBench, a benchmark to evaluate protein models' ability to predict and design high-affinity antibodies. We assess zero-shot correlation between model likelihoods and experimental binding affinities (Sec. 3.2) across curated datasets (Sec. 3.1), and validate generative performance by designing CDR-H3 variants that improve F045-092 binding to H1N1 influenza (Sec. 3.3.2).

## 3.1 DATASETS

We compiled 15 datasets of antigen, antibody heavy and light chain sequences, antibody structure, and experimental binding affinity measurements (Table 1).

For antibody structure, we focus on variable regions of heavy and light chain, where affinity conferring mutations occur. We only included datasets with at least 20 mutated antibodies to ensure the statistical significance of evaluation.

In each binding affinity study, antibody libraries are constructed through phage or yeast display, introducing mutations via deep mutational scanning or at targeted positions. Some libraries were designed computationally using machine learning Li et al. (2023); Shanehsazzadeh et al. (2023) or biophysical modeling Clark et al. (2006), resulting in variant sets ranging from 67 to 65,535. For consistency, we standardized experimental affinity measurements by taking the negative log for Kd and log for enrichment, so that higher values indicate stronger binding (Supplement 2, Table S1). While log enrichment is an indirect measure, it reflects how well a variant is retained after antigen-specific selection and correlates with binding strength. When normalized, it provides a scalable proxy for relative binding affinity in high-throughput screens. Notably, AbBiBench is designed to avoid data leakage: while wild-type antibodies and antigens may occur in public datasets such as OAS Olsen et al. (2022a) and SAbDab Dunbar et al. (2014), the specific antibody mutant–antigen complexes curated in our benchmark are not present in any known training corpus.

## 3.2 PROTEIN MODELS

Protein modeling is a fast-evolving active research area. We selected diverse pretrained protein and/or antibody models based on originality, code availability, and structure modality (Table S2).

**Masked Language Models:** Masked protein language models (MLMs) predict masked residues based on context and capture correlations between sequence motifs and higher-level functional properties, enabling their application in antibody design Hie et al. (2024); Meier et al. (2021). We evaluate several representative protein MLMs. `ESM-2` Lin et al. (2023) is trained on large-scale protein sequence datasets using a masked language modeling objective. `AntiBERTy` Ruffolo et al. (2021) is a 12-layer BERT model trained on 57 million heavy- and light-chain sequences from antibody database (OAS Olsen et al. (2022a)). Incorporating structural information into protein language models (PLMs) improves their ability to capture spatial context beyond sequence proximity. `SaProt` Su et al. (2023) extends `ESM-2` with structure-aware tokens from Foldseek Barrio-Hernandez et al. (2023); Van Kempen et al. (2024), embedding residue identity and local structure. `ProSST` Li et al. (2024) uses geometric vector perceptrons (GVP) encoder Jing et al. (2020) that discretizes local atomic neighborhoods into a compact codebook, with disentangled attention over sequence, structure, and position. `ESM-3` Hayes et al. (2025), an upgraded `ESM-2`, is a multimodal PLM that models sequence, structure, and function through discrete token tracks. Based on such general-purpose protein MLMs, several antibody-specific MLMs have been developed, including `CurrAb` Burbach & Briney (2025), `AbLang` Olsen et al. (2022b), and `IgBlend` Malherbe & Ucar (2024), to capture antibody-specific mutations driven by somatic recombination and hypermutation Ruffolo et al. (2021). `CurrAb`, a fine-tuned `ESM-2`, uses curriculum learning to gradually shift from unpaired to paired OAS Olsen et al. (2022a) antibody data while preserving pre-trained knowledge on general proteins.

**Autoregressive Protein Language Models:** Unlike MLMs, which predict masked tokens based on bidirectional context, autoregressive PLMs generate the next token using only left-to-right context. `ProGen2` Nijkamp et al. (2023) is a Transformer-decoder model that scales up to 6.4 B parameters and is trained on $\sim 1$ billion natural protein sequences. `ProtGPT2` Ferruz et al. (2022) adopts the `GPT-2` architecture with 738 M parameters and is trained end-to-end on $\sim 50$ million protein sequences. Inverse folding models are also autoregressive but based on global structure embeddings. `ProGen2` and `ProtGPT2` can serve as structure-agnostic comparisons for inverse folding models.

**Inverse Folding Models:** Inverse folding models aim to predict amino acid sequences from a given protein structure, often in an autoregressive manner. This approach has been applied to antibody mutant design by leveraging known Ab-Ag complex structures Shanker et al. (2024), enabling sequence exploration to identify mutations that preserve or enhance complex stability and binding affinity. Widely used models include `ProteinMPNN` Dauparas et al. (2022) and `ESM-IF` Hsu et al. (2022). `ProteinMPNN` uses a message-passing neural network Gilmer et al. (2017) to model residue interactions before autoregressive sequence generation. `ESM-IF1` combines a GVP encoder for extracting backbone-invariant features with a Transformer decoder. `AntiFold` Høie et al. (2025), based on `ESM-IF1`, is fine-tuned on antibody structures from SAbDab and OAS.

**Diffusion-Based Generative Models:** Diffusion models approach antibody design as a denoising process, transforming Gaussian noise into a target antibody by learning the joint distribution of atomic coordinates, orientations, and residue identities. `DiffAb` Luo et al. (2022) conditions on an antigen–antibody framework complex and jointly diffuses CDR sequence and structure. To assess backbone flexibility, we also evaluated fixed-backbone variants `DiffAb_fixbb`, a common setting in protein design Anishchenko et al. (2021); Hsu et al. (2022); Ingraham et al. (2019); Luo et al. (2022); Strokach et al. (2020); Tischer et al. (2020). `AbDiffuser` Martinkus et al. (2023) extends to full-atom generation with physical priors for side chains. `IgDiff` Cutting et al. (2025) performs *de novo* backbone generation by sampling variable-region backbones and then fills in the sequences using `AbMPNN` Dreyer et al. (2023).

**CDR Imputation in Geometric Representation:** Geometry-aware methods view affinity maturation as filling in missing CDRs within the explicit 3-D Ab-Ag interface. `MEAN` Kong et al. (2022) masks CDRs on an E(3)-equivariant residue–atom graph containing both chains and epitope; two alternating message-passing blocks jointly restore CDR sequence and backbone. `dyMEAN` Kong et al. (2023) upgrades this to full-atom, end-to-end design: conserved-framework initialization plus a "shadow paratope" lets the network emit paratope sequence, side-chain geometry, and binding pose in one

shot. We also considered fixed-backbone versions of `MEAN` and `dyMEAN`, dubbed `MEAN_fixbb` and `dyMEAN_fixbb`, respectively.

### 3.3 EVALUATION TASKS

Our benchmark comprises (i) zero-shot affinity prediction using retrospective experimental affinity data and (ii) antibody generation by sampling from the models.

#### 3.3.1 ZERO-SHOT PREDICTION OF EXPERIMENTAL BINDING AFFINITY USING MODEL LOG-LIKELIHOOD

To measure how well a model's zero-shot predicted log-likelihood aligns with wet-lab verified affinity, we calculated the Spearman correlation between the model likelihood and experimentally measured binding affinity. A high correlation indicates that the model assigns a higher likelihood to strong binders, suggesting that it can identify affinity-enhancing mutations in a zero-shot setting. To evaluate how effectively a model prioritizes the most promising antibodies, we also reported 5-fold precision@10 – the proportion of top 10 ranked variants that achieve at least 5-fold improvement in binding affinity compared to the wild type. The calculation of affinity fold change is detailed in Supplement 5. We harmonized the log-likelihood computation for all models under a unified setting: the input unit is the mutant-antigen complex, and the output is the likelihood of that complex. The details of likelihood computation across different types of models are provided in Supplement 4. In addition to the model's log likelihood, we report two biophysics-based affinity score as a baseline: binding free energy ($\Delta G$) and the relative solvent-accessible surface areas ($SASA$) of epitope residues (Sec. 3.3.2, Supplement 5). Lower values of both metrics imply stronger binding. To ensure consistent directionality with model log-likelihoods, we report $-\Delta G$ and $-SASA$.

#### 3.3.2 GENERATE ANTIBODY VARIANTS WITH STRONG H1N1 INFLUENZA AFFINITY

To assess whether protein models can generate antibody variants with improved binding to a specific antigen, we conducted a case study using F045-092 Ohshima et al. (2011), a naturally occurring antibody that targets the hemagglutinin (HA) protein of influenza H3N2. Notably, F045-092 fails to bind the H1N1 strain California2009 due to a steric barrier at the HA receptor-binding site Ekiert et al. (2012); Simmons et al. (2023). No experimentally determined structure of the F045-H1N1 complex exists in public databases, and this particular pair has not been studied in prior antibody design literature. As such, our study represents a completely novel setting, free from data leakage. The structure used in our study was computationally predicted using AlphaFold3. We used four representative models—`ESM-IF`, `SaProt`, `DiffAb`, and `MEAN`—to redesign the CDR-H3 loop of F045-092 while allowing up to five substitutions to maintain H3N2 cross-reactivity. We focus on generating mutations in the variable heavy chain (vH) due to its high diversity from V(D)J recombination and the central role of CDR-H3 in antigen binding Chen et al. (2024). The vH often acts as a unique antigen-specific signature Davies & Riechmann (1995), while the light chain often remains relatively conserved across functional antibodies Jaffe et al. (2022), making vH the most relevant region for affinity optimization. Each model generated 1,500 CDR-H3 variants from an input consisting of the masked F045-092 sequence and a predicted complex structure with the H1N1 HA1 protein (Supplement 7). Sampling strategies varied by model, including autoregressive prediction (`SaProt`), greedy heuristics (`ESM-IF`), and diffusion-based or graph-based generation (`DiffAb`, `MEAN`; see Supplement 7 for details).

We evaluated the generated antibody variants from two perspectives: sequence plausibility and binding potential. Sequence plausibility was assessed using the log-likelihood from `AntiBERTy` Ruffolo et al. (2021), reflecting how closely mutations align with natural antibody evolution. We also computed inverse folding likelihoods using `ProteinMPNN` to determine whether each sequence is compatible with its backbone structure—higher scores indicate greater foldability. Binding potential was evaluated using three structure-based metrics: (1) binding free energy ($\Delta G$) to estimate Ab–Ag interaction strength, (2) epitope SASA to measure differences in the solvent-accessible antigen surface area upon binding, and (3) complex structure confidence (AlphaFold pLDDT Abramson et al. (2024)) as a proxy for interface stability (Supplement 5).

To identify the final candidates, we used a two-phased screening approach (Fig. S1). In *Phase 1* we evaluated all 1,500 variants per model using sequence plausibility (`AntiBERTy` likelihood) and

$\Delta G$ – metrics that do not require structure prediction – and selected top 20% antibody variants. In *Phase 2*, we generated full Ab-Ag complex structures with AlphaFold 3 for this subset and computed pLDDT, epitope SASA, and inverse folding likelihood. We identified the final candidates as those in the Pareto-optimal set across all five metrics. We also examined the diversity of the selected antibody designs. We compared antibody PLM embeddings (`AntiBERTy`), sequence similarity (`cdrDist`), and structural deviation of the CDR-H3 loop (`cdrRMSD`) relative to the wild type (Supplement 5).

# 4 RESULTS

## 4.1 ZERO-SHOT PREDICTION OF EXPERIMENTAL BINDING AFFINITY USING MODEL LOG-LIKELIHOOD

As a result, inverse folding models achieved the highest accuracy in predicting experimental binding affinity, attaining the highest average Spearman correlation (Fig. 3) and highest 5-fold precision@10 (Fig. 4) for all inverse folding models tested. This high accuracy may stem from the model's broader structural scope. PLMs with local structure token have consistently performed best in general protein binding tasks Li et al. (2024); Su et al. (2023), but the same did not hold for antibodies. Inverse-folding models, which encode the entire Ab–Ag complex as a single global representation, consistently outperform models that rely on local structural tokens—such as `SaProt`, `ProSST`, and `ESM-3`. Besides, purely autoregressive sequence models without structure like `ProGen2` and `ProtGPT2` likewise fail to achieve competitive accuracy on the affinity prediction task. We shuffled the chain order of autoregressive models, but it did not increase the accuracy either (Table S3, Supplement 6). From a structural biology perspective, the inverse folding strategy is effective because protein function is ultimately determined by its three-dimensional structure, which is encoded by the underlying sequence. Mutations that maintain or improve structural integrity have a higher potential to enhance functional properties Shanker et al. (2024). On the other hand, by conditioning on the full antibody-antigen structure, inverse folding models can capture long-range residue interactions and contextual features at the binding interface—both of which are critical for affinity but often missed by models relying only on local sequence information Orlandi et al. (2020); Wang et al. (2018). Moreover,

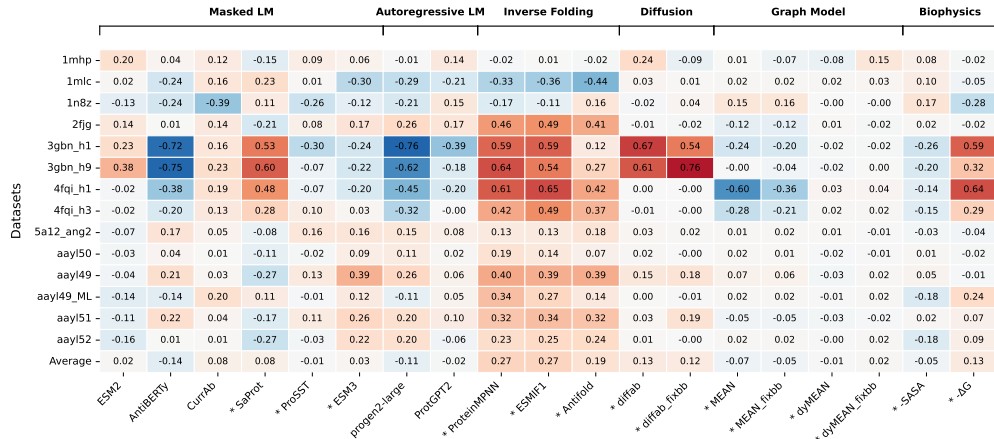

Figure 3: Spearman's rank correlation coefficients between model log likelihood from various protein models and experimental binding affinities across multiple datasets. Models marked with * are structure-informed.

among non–inverse folding models, SaProt which leverages local structural representations achieves the second-best performance, particularly in the 5-fold precision@10 metric. Interestingly, we also found that antibody-finetuned PLMs have varying impacts on zero-shot correlation to binding affinity. When we compare general PLM `ESM-2` to its Ab-finetuned model `CurrAb` Burbach & Briney (2025), `CurrAb` improved Spearman correlation by +0.074 across all datasets and precision@10 by +0.067. However, `AntiFold` Høie et al. (2025), which is an ESM-IF finetuned on structure of antibodies (OAS Olsen et al. (2022a)) and Ab-Ag complex (SAbDab Dunbar et al. (2014)) data,

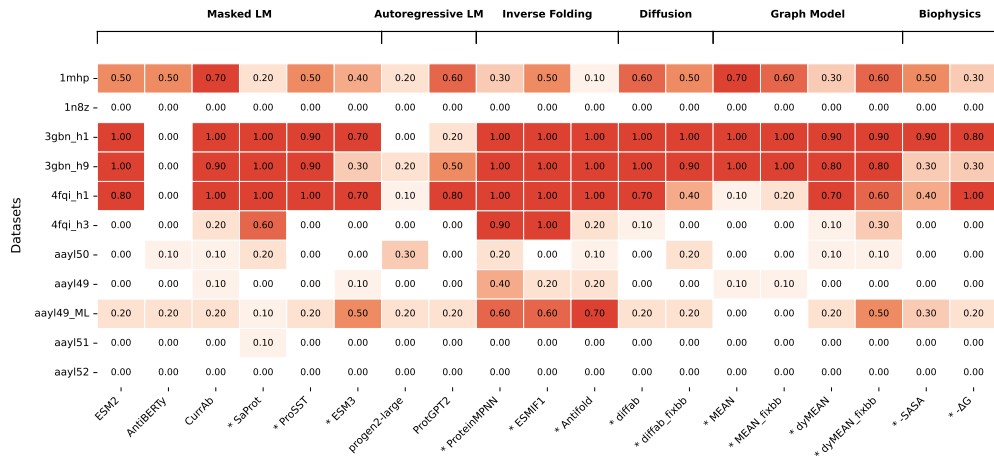

Figure 4: Proportion of top-10 ranked antibody designs achieving $\geq$5-fold affinity improvement across models and datasets. Only datasets reporting affinity as $-\log K_d$ were used. Datasets based on enrichment scores were excluded, as enrichment reflects relative sequence abundance and cannot determine fold change. Models marked with * are structure-informed.

deteriorated affinity correlation by -0.097 and precision@10 by -0.078 across all datasets, suggesting limited gains on Ab-specific finetuning and potential catastrophic forgetting.[3]

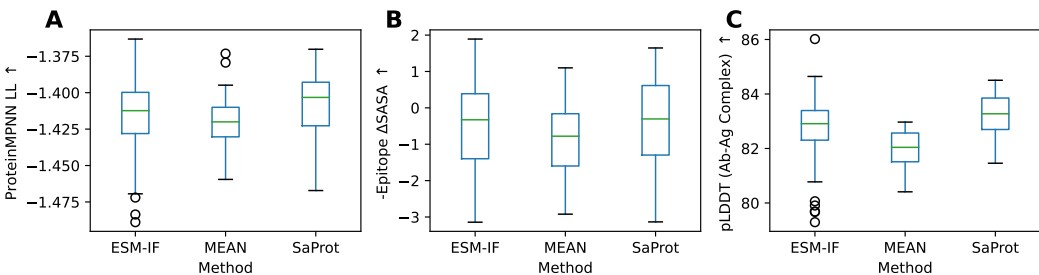

Figure 5: Binding potential and sequence plausibility in Phase 2. **A.** Sequence plausibility measured by ProteinMPNN likelihood. **B.** Binding potential measured by epitope SASA (presented as negative change, $-\Delta SASA$). **C.** Binding potential measured by pLDDT of Ab-Ag complex structure.

## 4.2 Generate Antibody Variants with Strong Affinity to H1N1 Influenza Virus

We evaluated whether selected models could generate antibody variants with stronger binding affinity to H1N1 than the wild-type F045-092. Each model produced 1,500 CDR-H3 variants with up to five mutations. As `DiffAb` often generated wild-type sequences with altered structures, we excluded duplicates, resulting in 467 valid `DiffAb` variants and a total of 4,967 unique variants across all models. In Phase 1 screening, `ESM-IF` and `SaProt` produced variants with both strong binding potential and high sequence plausibility. Their average $\Delta\Delta G$ values were -29.27 and -20.79, respectively, indicating substantial improvement in binding energy (Fig. S2, S3A). These variants also retained `AntiBERTy` plausibility scores (-0.663 for `ESM-IF` and -0.666 for `SaProt`) close to that of the wild type (-0.655; Fig. S3B). On the other hands, `DiffAb` generated plausible sequences (-0.657) but failed to improve binding energy ($\Delta\Delta G = 2.03$), suggesting mutations without affinity gain. In contrast, `MEAN` produced variants with improved binding ($\Delta\Delta G = -13.95$), but at the

---

[3]Note that `AntiFold` reports higher Spearman correlation with `1mlc` dataset (0.427) – see Fig S9 Høie et al. (2025)). This discrepancy occurs because `AntiFold` is evaluated using light and heavy chain variants measuring log-likelihood of the CDR regions, whereas AbBiBench excludes light chain mutants and report log-likelihood values of the entire Ab-Ag complex.

cost of lower sequence plausibility (-0.680), indicating a trade-off between biophysical fitness and evolutionary realism (Fig. S3). Among the top 20% of Phase 1 variants, 158, 91, and 26 candidates were selected from `ESM-IF`, `SaProt`, and `MEAN`, respectively. In Phase 2, `SaProt` achieved the highest sequence plausibility (ProteinMPNN log-likelihood = -1.406) and complex structure integrity (pLDDT = 83.22), followed by `ESM-IF` (-1.415, pLDDT = 82.82; Fig. 5, S4).

To assess the sequence diversity, we visualized CDR-H3 embeddings using AntiB-ERTy (Fig. S2B). `MEAN` variants formed a tight, isolated cluster, while `ESM-IF` and `SaProt` occupied broader, partially overlapping regions—indicating greater diversity.

Structural comparison revealed that `SaProt` generated CDR-H3 variants closest to wild type (mean cdrRMSD = 2.28), followed by `ESM-IF` (2.70) and `MEAN` (3.25). `SaProt` also produced variants with higher CDR-H3 structural confidence (pLDDT), suggesting better foldability (Fig. S7, S5). Sequence divergence analysis using cdrDist showed `SaProt` variants were most similar to wild type (0.222), compared to `ESM-IF` (0.279) and `MEAN` (0.277) (Fig. S7, S6). This aligns with the average number of substitutions per CDR-H3: 2.98 for `SaProt`, 3.03 for `MEAN`, and 3.75 for `ESM-IF`.

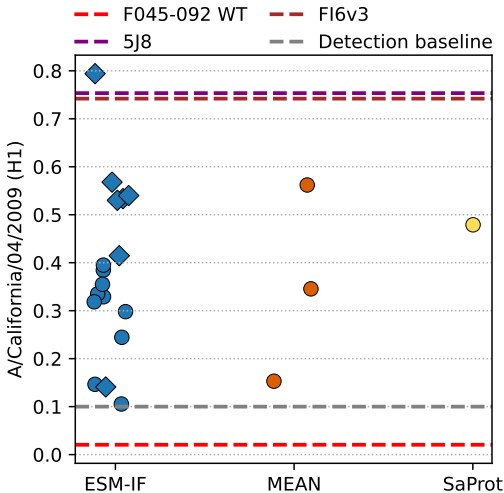

**In vitro experiments.** We performed ELISA binding assays on the 21 designed mutants against hemagglutinin (HA) from A/California/2009 (H1). The Figure 6 shows that all variants produced signals above the detection threshold for H1, indicating a gain of H1 binding activity. This suggests that our in-silico evaluation pipeline enables efficient screening of affinity-enhancing mutations from multiple models, thereby reducing the time and cost associated with experimental validation. Notably, the greatest gain in affinity against the H1 subtype was achieved by the ESM-IF derived designs supporting the conclusion of our in silico evaluation.

Figure 6: ELISA $OD_{450}$ signals for 21 model-designed F045-092 mutants and controls against H1 hemagglutinin. Three antibodies were used as controls, including the wild-type F045-092 (H3-specific binder), 5J8 (H1-specific binder), and FI6v3 (a broadly neutralizing antibody for both subtypes). Diamonds represent mutants with higher $OD_{450}$ values than F045-092 against H3.

## 5 CONCLUSION AND LIMITATIONS

This study introduces AbBiBench, a biologically relevant and structurally informed benchmarking framework for antibody binding affinity maturation and optimization. Recognizing the limitations of traditional computational evaluation metrics, our approach explicitly incorporates antibody-antigen complex information, thereby aligning more closely with the biological realities of antibody interactions. Our results demonstrate that global structure-informed protein language model used for inverse folding methods, such as `ESM-IF` and `ProteinMPNN`, outperform other evaluated computational models, primarily due to their effective integration of structural context. In a case study focused on redesigning the F045-092 antibody for binding to the H1N1 influenza subtype, we identified 21 Pareto-optimal antibody variants with improved predicted affinity and structural integrity. These 21 variants have been successfully expressed in vitro, and conduct ELISA assays to quantify their binding affinity to H1N1 hemagglutinin. The results of these assays directly validate the computational predictions and help assess the true binding potential of the designed antibodies. Limitations of this study include the lack of currently available experimental neutralization data and the relatively small size of some benchmark datasets (Supplement 8). In future work, we plan to expand the benchmark to include additional therapeutic properties such as stability, immunogenicity, and developability, and to incorporate functional assay data (e.g., IC50) to further align model evaluations with biological outcomes.

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

## SUPPLEMENT MATERIALS

### CONTENTS

## 1   CODE, DATASET, AND COMPUTATIONAL RESOURCES

Code repository and leaderboard are available in `https://github.com/`
`MSBMI-SAFE/AbBiBench`. The benchmarking dataset is available in
https://huggingface.co/datasets/AbBibench/Antibody_Binding_Benchmark_Dataset. Model
training is not required in this study. All inference tasks are conducted on a single NVIDIA H100
80GB GPU per model.

```python
import requests
from huggingface_hub.file_download import build_hf_headers
from mlcroissant import Dataset

# Login using e.g. `huggingface-cli login` to access this dataset
headers = build_hf_headers()  # handles authentication
jsonld = requests.get(
    "https://huggingface.co/api/datasets/AbBibench/
        Antibody_Binding_Benchmark_Dataset/croissant",
    headers=headers
).json()
ds = Dataset(jsonld=jsonld)
records = ds.records("default")
```

Listing 1: Accessing the AbBibench dataset using Croissant

## 2   BINDING AFFINITY DATA DETAILS

To obtain a robust experimental baseline for AbBiBench, we collated 16 binding-affinity assays drawn
from open-source studies. Selection was guided by two criteria: (i) maximizing Ab-Ag diversity
and (ii) having enough mutants per assay to ensure statistical power of our correlation analyses. For
each dataset we transformed the reported binding metric to common scales and computed Spearman
correlation between experimental binding scores and model log-likelihoods, providing a zero-shot
test of each model's ability to recognize affinity-improving mutations. The following sections outline,
per study how raw measurements were curated and converted into the benchmark scores.

**Influenza data**   For our benchmarking study, we derived data (processed by Shanker et al. (2024))
from an experiment investigating the binding affinity landscapes of two broadly neutralizing anti-
influenza antibodies (bnAbs), CR6261 (PDB ID: 3GBN) and CR9114 (PDB ID: 4FQI) Phillips et al.
(2021). Combinatorially complete libraries of all evolutionary intermediates were constructed for
each bnAb's heavy chain, spanning 11 mutations for CR6261 and 16 for CR9114. After removing
entries missing dissociation constant $K_d$ values, we used $-log(K_d)$ (in M) for benchmarking. This
yielded 1,887 data points for the H1N1 subtype and 1,842 for the H9H2 subtype (both against
CR6261), as well as 65,094 H1N1 and 65,535 H3N2 data points (both against CR9114).

| ID | Chains | Chain lengths | Library selection | Mutated regions | Study |
|---|---|---|---|---|---|
| 2fjg | H/L/V | 118/107/95 | Phage display DMS | both | Koenig et al. (2017) |
| 3gbn_h1 | H/L/A,B | 121/109/328/173 | Germline-reverted (yeast display) | both | Phillips et al. (2021) |
| 3gbn_h9 | H/L/A,B | 121/109/328/173 | Germline-reverted (yeast display) | both | Phillips et al. (2021) |
| 4fqi_h1 | H/L/A,B | 121/109/324/176 | Germline-reverted (yeast display) | both | Phillips et al. (2021) |
| 4fqi_h3 | H/L/A,B | 121/109/324/176 | Germline-reverted (yeast display) | both | Phillips et al. (2021) |
| 1mlc | B/A/E | 117/109/129 | Yeast display DMS | both | Warszawski et al. (2020) |
| aayl49 | B/C/A | 118/113/14 | Phage display DMS | CDRs | Engelhart et al. (2022) |
| aayl49_ML | B/C/A | 118/113/14 | ML-generated scFv (phage display) | CDRs | Li et al. (2023) |
| aayl51 | B/C/A | 119/115/14 | Phage display DMS | CDRs | Engelhart et al. (2022) |
| 1n8z | B/A/C | 121/109/581 | Zero-shot generative AI model | CDRs | Shanehsazzadeh et al. (2023) |
| 1mhp | H/L/A | 118/107/184 | Mutant library (in-silico biophysics) | both | Clark et al. (2006) |
| 4d6_her2 | B/A/C | 121/109/581 | Yeast display DMS | CDRs | Minot & Reddy (2024) |
| 5a12_ang2 | H/L/A | 215/213/220 | Yeast display DMS | CDRs | Minot & Reddy (2024) |
| 5a12_vegf | H/L/D | 211/213/99 | Yeast display DMS | CDRs | Minot & Reddy (2024) |

Table S1: Dataset metadata (extended version of Table 1): "Chains" lists chain IDs in the PDB entry, "Chain lengths" are the corresponding chain sequence lengths in the same order; "Mutated regions" indicates whether the experimental assays span both frameworks and CDRs or are restricted to CDRs only.

**VEGF data** Mutational data for anti-VEGF antibody was sourced from two different studies. We collected data from Shanker et al. (2024), which reported deep mutational scanning results from Koenig et al. (2017) to systematically analyze the impact of antibody mutations remote to the antigen-binding site. For benchmarking, we derived mutational data from positions 2–113 of the variable heavy-chain region (Kabat numbering), obtained from the G6.31 Fab phage display study for the models' input along with the structure (PDB ID: 2FJG). We collected the corresponding binding enrichment ratios (ER), defined as the log, frequency ratio of each mutation post-selection relative to pre-selection. Here, higher ER values indicate mutations having enhanced fitness, primarily reflecting increased binding affinity toward the VEGF antigen. The resulting dataset contained 2,223 heavy chain variants.

In a second dataset (ID: 5a12_vegf), Minot & Reddy (2024) employed yeast-display mutagenesis libraries and fluorescence-activated cell sorting (FACS) to generate training data for machine-guided antibody engineering. Here, variant fitness was quantified as the wild-type-normalized log enrichment between the selected and unselected libraries. We obtained these values in processed form from BindingGYM, which replicated the original workflow and standardized the calculation as a fitness score. This resulted in a total of 29,990 variants across the heavy and light chains.

**SARS-CoV-2 data** This dataset comprises binding assay scores of human antibodies targeting the HR2 region of the SARS-CoV-2 peptide (spike protein) developed for the purpose of benchmarking machine learning models Engelhart et al. (2022). Through phage display, three antibodies were identified as binders from which the antibody library was designed by making up to k=3 point mutations in the CDR regions. Our benchmarking study focuses on two mutated variable heavy chain sequences (AAYL49 and AAYL51 assays) and two mutated variable light chain sequences (AAYL50 and AAYL52). Disassociation constant (Kd) values were provided in nM which was converted to M to derive the -log (Kd) values used for correlation (keeping consistency with Shanker et al. (2024)). After averaging triplicate results, removing negative controls and non-heavy chain related assays we collated 4,312 datapoints from AAYL49, 4,320 from AAYL51, 11,473 from AAYL50, and 13,324 from AAYL52.

An extension to this study involved utilizing the previously described SARS-CoV-2 data to train a Bayesian language model for scFv design. The AAYL49 assay-trained model was used to generate scFv libraries enriched for improved affinity. From these libraries, we specifically extracted mutations introduced in the heavy-chain variable region that were not present in the phage display-derived AAYL49 library, resulting in dataset AAYL49_ML of 8,953 mutated heavy chain sequences. Due to the lack of structural information from these studies, we used AlphaFold 3 Abramson et al. (2024) to generate the antibody-antigen complexes for the SARS-CoV-2 data derived from this study.

**Lysozyme data** Similar to the anti-VEGF antibody study, this study investigates the mutational tolerance of a variable fragment of an anti-lysozyme antibody Warszawski et al. (2020). A deep mutational scanning approach was applied to a yeast-display library, where single-point mutations were introduced at 68 positions of the heavy-chain variable region (PDB ID: 1MLC), each mutated to all 20 standard amino acids in a combinatorial matrix fashion. The resulting mutation data were converted into tabular format by reconstructing the mutated heavy-chain sequences and noting their corresponding log-enrichment ratios. For benchmarking, we utilized this mutational tolerance map, which provided a dataset containing a total of 1,297 ER values.

**HER2 data** The anti-HER2 data was sourced from two studies. The first aimed to develop generative AI models capable of producing antibody binders without iterative optimization (zero-shot generation) Shanehsazzadeh et al. (2023). The authors demonstrated their approach by computationally designing the heavy-chain complementarity-determining region 3 (HCDR3) of the HER2-targeting antibody trastuzumab (PDB ID: 1N8Z). For our correlation analysis, we filtered this dataset to include only mutated sequences with an HCDR3 length matching the wild-type trastuzumab sequence. The reported binding affinity (Kd) values, originally measured in nanomolar (nM), were converted to molar (M), and subsequently transformed to negative log-scale $-log(K_d)$. In total, 419 mutated HCDR3 sequences from this dataset were utilized for our benchmarking study.

From Minot & Reddy (2024) we obtained an additional dataset (ID: 4d5_her2), curated by Binding-GYM as described above, comprising 2,080 heavy- and light-chain variants.

**Integrin-$\alpha$-1 data**  In this study from 2006, researchers sought to improve the binding affinity of the AQC2 antibody fragment to the I-domain of integrin VLA1 Clark et al. (2006). Their approach involved utilizing structure-based computational methods to propose mutants through side chain repacking Hanf (2002); Looger & Hellinga (2001); Wisz & Hellinga (2003) and electrostatic optimization Kangas & Tidor (1998); MacKerell et al. (1998). Libraries were computationally generated by varying nearly all antigen-contacting residue positions in both antibody heavy and light chains (PDB ID: 1MHP). Successful single mutations identified experimentally were then combined to further increase affinity. Binding affinity ($K_d^{\mathrm{mut}}$) was estimated as the fold change affinity relative to wildtype using a competition ELISA. This resulted in a total of 67 variants with mutations in either or both chains.

**Ang2**  This dataset (ID: 5a12_ang2), sourced from Minot & Reddy (2024) and processed by BindingGYM following the same workflow described earlier, contained 943 variants across the heavy and light chains.

## 3 COMPARISON OF PROTEIN MODELS

We systematically compare the baseline methods based on the type of proteins and structure modality in training data. All models use sequence information as training data (Table S2).

| Category | Model | Training data | | Structure modality | |
|---|---|---|---|---|---|
| | | **Protein** | **Antibody** | **Local** | **Global** |
| Masked PLM | ESM2 Lin et al. (2023) | x | | | |
| | ESM3 Hayes et al. (2025) | x | | x | x |
| | SaProt Su et al. (2023) | x | | x | |
| | ProSST Li et al. (2024) | x | | x | x |
| | AntiBERTy Ruffolo et al. (2021) | | x | | |
| | CurrAb Burbach & Briney (2025) | x | x | | |
| Autoregressive PLM | ProGen2 Nijkamp et al. (2023) | x | | | |
| | ProtGPT-2 Ferruz et al. (2022) | x | | | |
| Inverse folding | ProteinMPNN Dauparas et al. (2022) | x | | | x |
| | ESM-IF Hsu et al. (2022) | x | | | x |
| | AntiFold Høie et al. (2025) | x | x | | x |
| Diffusion-based generative models | Diffab Luo et al. (2022) | | x | | x |
| | Diffab_fixbb Luo et al. (2022) | | x | | x |
| CDR imputation in geometric | MEAN Kong et al. (2022) | | x | | x |
| | MEAN_fixbb Kong et al. (2022) | | x | | x |
| | dyMEAN Kong et al. (2023) | | x | | x |
| | dyMEAN_fixbb Kong et al. (2023) | | x | | x |

Table S2: Comparison of protein modeling methods. PLM: Protein language model.

## 4 CALCULATION DETAILS OF LOG-LIKELIHOOD FOR DIFFERENT MODELS

Algorithm 1 illustrates the process of calculating the Spearman correlation ($\rho$) between model log-likelihoods and measured affinities for each model. Following that, we describe the computation of zero-shot log-likelihoods across four model families: masked language models, inverse folding models, diffusion-based generative models, and graph-based CDR imputation models.

NOTATION

Let:

- $\mathbf{s} = (s_1, \ldots, s_N) \in \mathcal{A}^N$: full amino acid sequence of the antibody–antigen complex, where $\mathcal{A}$ is the amino acid vocabulary and $N$ is the total number of residues.

- $\mathbf{X} \in \mathbb{R}^{N \times 3}$: 3D backbone C$\alpha$ atom coordinates for all $N$ residues in the complex.

- $\mathbf{O} = (O_1, \ldots, O_N)$, where $O_i \in \mathrm{SO}(3)$: local 3D orientation of residue $i$.

---

**Algorithm 1** Compute the Spearman correlation ($\rho$) between model log-likelihoods and measured affinities

---

**Require:** Dataset $\mathcal{D} = \{(\mathbf{s}_i^{\mathrm{ab}}, \mathbf{s}_i^{\mathrm{ag}}, x_i, y_i)\}_{i=1}^M$, where:

- $\mathbf{s}_i^{\mathrm{ab}}$: antibody sequence of sample $i$
- $\mathbf{s}_i^{\mathrm{ag}}$: antigen sequence of sample $i$
- $\mathbf{z}_i$: structure information of the antibody–antigen complex; this may include atomic coordinates or orientation, depending on the model type
- $y_i$: experimentally measured binding affinity

**Ensure:** Spearman correlation $\rho$ between model log-likelihoods and experimental affinities
1: $\mathcal{L} \leftarrow [\,]$ {Initialize empty list for log-likelihoods}
2: **for** each $(\mathbf{s}_i^{\mathrm{ab}}, \mathbf{s}_i^{\mathrm{ag}}, \mathbf{z}_i, y_i) \in \mathcal{D}$ **do**
3:    $\ell_i \leftarrow \mathcal{M}.\textsc{LogLikelihood}(\mathbf{s}_i^{\mathrm{ab}}, \mathbf{s}_i^{\mathrm{ag}}, \mathbf{z}_i)$ {Omit $\mathbf{z}_i$ if $\mathcal{M}$ is sequence-only}
4:    Append $\ell_i$ to $\mathcal{L}$
5: **end for**
6: $\rho \leftarrow \textsc{SpearmanCorr}(\mathcal{L}, \{y_i\}_{i=1}^M)$
7: **return** $(\mathcal{L}, \rho)$

---

- $\mathcal{C} \subset \{1, \dots, N_{\mathrm{CDR}}\}$: set of indices corresponding to CDR (complementarity-determining region) residues.
- $h_i \in \mathbb{R}^d$: node embedding for residue $i$, typically obtained from a GNN or transformer encoder.
- $Z_i, \hat{Z}_i \in \mathbb{R}^{3 \times k}$: predicted and ground-truth coordinates of $k$ backbone or side-chain atoms of residue $i$. For MEAN, $k = 4$ corresponds to backbone atoms N, C$\alpha$, C, and O; dyMEAN extends this to up to 14 atoms including side-chain atoms.
- Dynamic design graph $\mathbf{G} = (\mathcal{V}, \mathcal{E})$: a residue-level graph over the full antibody–antigen complex used in co-design models. Each node $v_i \in \mathcal{V}$ has embedding $h_i$ and structure $Z_i$. For $i \in \mathcal{C}$, features and coordinates are masked and updated during message passing.
- Structure-frozen graph $\mathbf{G}_{\mathrm{fix}}$: the graph used in structure-fixed design (fixbb) settings. Residues outside $\mathcal{C}$ provide fixed sequence and structural context, while residues in $\mathcal{C}$ are masked in sequence but retain their fixed structure during prediction.

### 4.1 MASKED LANGUAGE MODELS

To calculate the likelihood of a sequence $\mathbf{s}$ with masked language models, we approximate the log-likelihood by summing the log-probabilities of each residue in the unmasked sequence:

- For structure-aware MLMs:

$$\log P(\mathbf{s}) = \sum_{i=1}^N \log P(s_i \mid \mathbf{s}, \mathbf{X})$$

- For structure-agnostic MLMs:

$$\log P(\mathbf{s}) = \sum_{i=1}^N \log P(s_i \mid \mathbf{s})$$

This method uses the full, unmasked sequence as input. Although this approach does not align with the model's training objective, it serves as an efficient approximation for estimating likelihood Johnson et al. (2025).

### 4.2 INVERSE FOLDING MODELS

Inverse folding models condition on the full backbone structure and autoregressively predict the sequence:

$$\log P(\mathbf{s} \mid \mathbf{X}) = \sum_{i=1}^N \log P(s_i \mid \mathbf{s}_{<i}, \mathbf{X})$$

where $\mathbf{s}_{<i}$ is the prefix up to position $i-1$.

## 4.3 DIFFUSION-BASED GENERATIVE MODELS

Diffusion models learn a denoising process in the CDR region, jointly modeling its sequence and structure. This generation process is conditioned on the structure context, which includes coordinates of backbone atoms, N, $C_\alpha$, C, and O, and orientations of side-chain atom, $C_\beta$. Let the set of CDR residues be:

$$\mathcal{R} = \{(s_j, x_j, O_j) \mid j \in \mathcal{C}\}$$

The conditioning context includes all other residues:

$$\mathbf{C} = \{(s_i, x_i, O_i) \mid i \notin \mathcal{C}\}$$

The objective for sequences for CDR residues are

$$\mathcal{L}_t^{\text{type}} = \mathbb{E}_{R_t \sim p} \left[ \frac{1}{m} \sum_{j=1}^{m} D_{\text{KL}} \left( q(s_{t-1}^j \mid s_t^j, s_0^j) \,\|\, p_\theta(s_{t-1}^j \mid R_t, C) \right) \right],$$

where $q(\cdot)$ denotes the *forward* (noising) process, which gradually perturbs the CDR input over $T$ steps. The function $p_\theta(\cdot)$ represents the learned *reverse* (denoising) process, which predicts how to revert the noise at each step, conditioned on the context $\mathbf{C}$. The model is trained to denoise the CDR residues from increasingly corrupted inputs, ensuring that the generated sequences remain consistent with the surrounding structural and sequence context.

In a similar manner, the objective for generating $C_\alpha$ coordinates is defined as:

$$\mathcal{L}_t^{\text{pos}} = \mathbb{E} \left[ \frac{1}{m} \sum_{j=1}^{m} \|\boldsymbol{\epsilon}_j - G(R_t, \mathbf{C})\|_2^2 \right],$$

where $G(\cdot)$ is a neural network trained to predict the standard Gaussian noise added during the forward diffusion process.

In addition, orientation is also modeled within the diffusion framework using the following objective:

$$\mathcal{L}_t^{\text{ori}} = \mathbb{E} \left[ \frac{1}{m} \sum_{j=1}^{m} \left\| \mathbf{O}_0^{j\top} \hat{\mathbf{O}}_{t-1}^j - \mathbf{I} \right\|_F^2 \right]$$

where $\mathbf{O}_0^j \in \mathbb{R}^{3\times3}$ denotes the ground-truth rotation matrix for residue $j$ at timestep 0, $\hat{\mathbf{O}}_{t-1}^j \in \mathbb{R}^{3\times3}$ is the predicted rotation matrix at timestep $t-1$, $\mathbf{I} \in \mathbb{R}^{3\times3}$ is the identity matrix, and $\|\cdot\|_F$ denotes the Frobenius norm.

Finally, the overall training objective is formulated as:

$$L = \mathbb{E}_{t \sim \text{Uniform}(1...T)} \left[ \mathcal{L}_t^{\text{type}} + \mathcal{L}_t^{\text{pos}} + \mathcal{L}_t^{\text{ori}} \right]$$

where the total loss at each timestep combines the type prediction loss, positional loss, and orientation loss. Further details regarding the model architecture and training procedure can be found in the original publication Luo et al. (2022).

**Structure-Fixed Variant (`DiffAb_fixbb`)**  To isolate sequence-level generation, we define a structure-frozen variant in which all coordinates and orientations within the context are fixed. In this case, only the sequence of the CDR region is masked and excluded from the context. Consequently, we only utilize the sequence loss $\mathcal{L}_t^{\text{type}}$, while the position and orientation of the CDR residues remain fixed.

## 4.4 GRAPH-BASED CDR IMPUTATION MODELS

Graph-based antibody design models such as MEAN and dyMEAN treat the antibody–antigen complex as a spatially structured graph and aim to jointly predict the amino acid sequence and full-atom structure (backbone + sidechains) of masked CDR regions. These models are built upon E(3)-equivariant graph neural networks, ensuring that predictions are consistent under rotation and translation.

Let the antibody–antigen complex be represented as a graph $\mathbf{G} = (\mathcal{V}, \mathcal{E})$, where each node $v_i \in \mathcal{V}$ corresponds to a residue with a feature embedding $h_i$ and a full-atom coordinate matrix $Z_i \in \mathbb{R}^{3 \times c_i}$, where $c_i$ is the number of atoms (varies across residue types). A subset of nodes $\mathcal{C} \subset \mathcal{V}$ corresponds to masked CDR residues for which both identity and structure are to be generated.

The model iteratively updates both $h_i$ and $Z_i$ using multi-channel equivariant message passing:

$$\{h_i^{(t+1)}, Z_i^{(t+1)}\}_{i \in \mathcal{V}} = \text{GNN}_\theta(\{h_i^{(t)}, Z_i^{(t)}\}_{i \in \mathcal{V}}, \mathbf{G})$$

The amino acid type for residue $i \in \mathcal{C}$ is predicted from the final hidden representation:

$$p_i = \text{Softmax}(W h_i^{(T)})$$

and the full-atom coordinates are given directly as $Z_i^{(T)}$.

The training loss consists of both sequence and structure terms:

$$L = \sum_{i \in \mathcal{C}} \left[ l_{ce}(p_i, \hat{p}_i) + \lambda l_{huber}(Z_i, \hat{Z}_i) \right],$$

where $\hat{p}_i$ and $\hat{Z}_i$ represent the ground-truth atom sequence distribution and coordinates, with $l_{ce}$ and $l_{huber}$ corresponding to the cross-entropy loss and Huber loss for sequence and coordinates, respectively.

**Structure-Fixed Variants (`MEAN_fixbb`, `dyMEAN_fixbb`)**  To allow comparison with fixed-backbone models such as inverse folding, we define structure-frozen variants that predict sequence identities only, conditioned on a fixed geometry graph $\mathbf{G}_{\text{fix}}$. These models retain the same message-passing architecture but discard the coordinate regression loss. Their objective is reduced to:

$$L = \sum_{i \in \mathcal{C}} l_{ce}(p_i, \hat{p}_i),$$

These models perform conditional full-atom sequence design under strict geometric constraints and are especially useful for evaluating structure-aware sequence recovery in antibody design.

## 5 COMPUTATIONAL EVALUATION METRICS

- **Binding Energy** FoldX is a computational force field that calculates the free energy of protein-protein interactions by evaluating multiple physical energy terms. These terms include van der Waals forces between atoms, both inter- and intra-molecular hydrogen bonding, electrostatic interactions between charged groups, and additional contributions from solvation and entropy Guerois et al. (2002). The algorithm is widely used to predict the impact of mutations on protein stability, defined as the difference in Gibbs free energy ($\Delta\Delta G$) between mutant and wild-type proteins ($\Delta G_{\text{variant}} - \Delta G_{\text{wild type}}$). In this study, we used FoldX's `analyseComplexChains` command to quantify the binding energy differences between antigen and antibody chains within their molecular complex. A lower value of $\Delta\Delta G$ indicates a stronger binding upon mutation.

- **Epitope SASA** Solvent Accessible Surface Area (SASA) is a computational measure that quantifies the exposure of protein residues to the surrounding solvent. Using the FreeSASA Python module Mitternacht (2016), we calculated the surface accessibility of epitope residues in both wild-type and mutant antibody-antigen complexes. A decrease in solvent accessible surface area typically indicates tighter packing at the antibody-antigen interface, which often correlates with stronger binding affinity. We defined the epitope as antigen residues located within 5 Å of the antibody chain, as this distance threshold effectively captures the antibody-antigen binding interface Abramson et al. (2024); Myung et al. (2023). We employed relative SASA values to enable meaningful comparisons across different protein structures, representing the ratio of actual surface area to the maximum possible surface area for each residue type. The overall change in epitope accessibility was quantified as:

$$\Delta\text{SASA} = \sum \left(\text{relative SASA}_{\text{variant}}\right) - \sum \left(\text{relative SASA}_{\text{wild type}}\right),$$

where the summation is performed over all epitope residues.

- **cdrDist** Following the approach proposed by Thakkar and Bailey-Kellogg Thakkar & Bailey-Kellogg (2019), we compute the sequence distance using the normalized Smith–Waterman alignment score. Let $S_{\text{wild type}}$ denote the wildtype CDR-H3 sequence and $S_{\text{variant}}$ the mutant CDR-H3 sequence. The distance between two sequences is defined as:

$$\text{CDRdist}(S_{\text{wild type}}, S_{\text{variant}}) = 1 - \frac{\text{SW}(S_{\text{wild type}}, S_{\text{variant}})^2}{\text{SW}(S_{\text{wild type}}, S_{\text{wild type}}) \cdot \text{SW}(S_{\text{variant}}, S_{\text{variant}})}$$

where $\text{SW}(X, Y)$ denotes the Smith–Waterman local alignment score between sequences $X$ and $Y$. This formulation penalizes dissimilar alignments more heavily and ensures that the distance is normalized with respect to the self-alignment scores of the sequences being compared. The distance lies in the interval $[0, 1]$, where 0 indicates identical sequences.

- **cdrRMSD** We used the Kabsch algorithm to superimpose the $C_\alpha$ atoms of the residues comprising each sampled CDR-H3 loop onto the corresponding CDR-H3 region of the wild-type structure, and calculated the resulting root-mean-square deviation (RMSD).

- **Affinity Fold Change** The fold change was calculated by taking the difference between the $\text{pK}_d$ values of the mutant and wild-type antibodies (i.e., $-\log_{10} K_d$), and exponentiating the result as $10^{(\text{p}K_{d,\,\text{mutant}} - \text{p}K_{d,\,\text{wild-type}})}$. This yields the ratio of the dissociation constants $K_d$ between the wild-type and mutant antibodies.

## 6 Impact of Chain Order in Autoregressive PLMs in Correlation Studies

Since autoregressive models (ESM-IF1 Hsu et al. (2022), ProGen Nijkamp et al. (2023), ProGPT2 Ferruz et al. (2022)) use previous tokens as context to predict the next, it was hypothesized that providing antigen (mimicking SHM) and light chain context before the mutated heavy chain could improve zero-shot correlation of general autoregressive PLMs to experimental binding affinity. The benchmarking results show otherwise, with sporadic and minimal impact of chain ordering seen across both structure-based (ESM-IF1 Hsu et al. (2022)) and sequence-based (ProGen2 Nijkamp et al. (2023), ProGPT 2 Ferruz et al. (2022)) correlation studies across the eleven Ab-Ag datasets. Therefore, demonstrating that autoregressive PLMs trained on single chain protein data remain largely insensitive to multimer chain ordering.

| Datasets
Models | anti-VEGF
2fjg | Influenza
3gbn_h1 | Influenza
3gbn_h9 | Influenza
4fqi_h1 | Influenza
4fqi_h3 | SARS-CoV-2
AAYL49 | SARS-CoV-2
AAYL51 | SARS-CoV-2
AAYL49(ML) | anti-lysozyme
1mlc | anti-HER2
1n8z | anti-integrin
1mhp |
|---|---|---|---|---|---|---|---|---|---|---|---|
| **ESM-IF** | *0.5504* | *0.5950* | *0.5399* | **0.6459** | *0.4938* | *0.3871* | *0.3439* | *0.2662* | *-0.3574* | *-0.1083* | *-0.3564* |
| **ESM-IF (A/H/L)** | 0.5586 | **0.6015** | **0.5399** | 0.5306 | 0.4745 | **0.3958** | 0.3335 | **0.2880** | -0.3569 | -0.2180 | -0.3053* |
| **ESM-IF (L/A/H)** | **0.5600** | 0.2408 | 0.2606 | 0.6286 | 0.3855 | 0.3283 | 0.3694 | 0.2694 | -0.3578 | -0.0931* | -0.3904 |
| **ProtGPT2** | 0.0372 | -0.3913 | -0.1763 | -0.2017 | -0.0014* | 0.0634 | 0.1028 | 0.0634 | -0.2120* | 0.1463 | -0.0962* |
| **ProtGPT2 (A/H/L)** | 0.0114* | -0.4254 | -0.4622 | -0.2507 | 0.0246* | 0.0783 | 0.0992 | -0.1617 | **0.1489** | **0.1834*** |
| **ProtGPT2 (L/A/H)** | -0.0536 | -0.5087 | -0.3812 | -0.4656 | -0.2340 | 0.0858 | 0.0903 | 0.0883 | -0.2142 | -0.0325* | -0.2441* |
| **ProGen2 (Base)** | *0.2861* | *-0.6707* | *-0.5972* | *-0.4643* | *-0.3127* | *0.2668* | *0.1934* | *-0.1124* | *-0.3851* | *-0.1932* | *-0.3532* |
| **ProGen2 (Base) (A/H/L)** | 0.3554 | -0.5716 | -0.5594 | -0.4156 | -0.2504 | 0.2665 | 0.2001 | -0.0865 | -0.2985 | -0.1209 | -0.1503* |
| **ProGen2 (Base) (L/A/H)** | 0.4324 | -0.5417 | -0.5498 | -0.3890 | -0.2339 | 0.2537 | 0.2501 | -0.1122 | -0.3165 | -0.0324* | -0.1746* |
| **ProGen2 (Small)** | *0.2039* | *-0.6800* | *0.6498* | *-0.6321* | *-0.3592* | *0.2814* | *0.2294* | *-0.0412* | *-0.2222* | *-0.2232* | *-0.1045* |
| **ProGen2 (Small) (A/H/L)** | 0.2914 | -0.4463 | -0.4975 | -0.5841 | -0.3278 | 0.2812 | 0.2103 | -0.0417 | -0.2340 | -0.1180 | -0.3592 |
| **ProGen2 (Small) (L/A/H)** | 0.3851 | -0.4762 | -0.4762 | -0.5632 | -0.3318 | 0.3014 | 0.2243 | -0.0899 | -0.2646 | -0.2036 | -0.0534* |
| **ProGen2 (Medium)** | *0.2537* | *-0.6887* | *-0.6204* | *-0.5284* | *-0.3729* | *0.2865* | *0.2117* | *-0.0702* | *-0.2780* | *-0.2669* | *-0.0221* |
| **ProGen2 (Medium) (A/H/L)** | 0.3023 | -0.4679 | -0.5396 | -0.6204 | -0.4178 | 0.2980 | 0.2215 | -0.0996 | -0.2847 | -0.0780* | 0.1231* |
| **ProGen2 (Medium) (L/A/H)** | 0.3710 | -0.4543 | -0.5496 | -0.6400 | -0.4292 | 0.3069 | 0.2005 | -0.0565 | -0.2962 | 0.0046* | 0.1162* |
| **ProGen2 (Large)** | *0.2704* | *-0.7555* | *-0.6245* | *-0.4478* | *-0.3203* | *0.2578* | *0.2017* | *-0.1124* | *-0.2869* | *-0.2086* | *-0.3777* |
| **ProGen2 (Large) (A/H/L)** | 0.3229 | -0.6290 | -0.6516 | -0.4859 | -0.3789 | 0.2764 | 0.2239 | -0.1038 | -0.2693 | 0.0270* | -0.1325* |
| **ProGen2 (Large) (L/A/H)** | 0.3097 | -0.6518 | -0.6661 | -0.4533 | -0.3625 | 0.2750 | 0.2109 | -0.0512 | -0.3391 | 0.0499* | -0.0344* |

Table S3: Performance of autoregressive models across diverse Ab-Ag binding assays. Highest correlation per dataset is marked in bold and baseline model values are italicized. Asterisk (*) denotes correlation values are insignificant (p-value ¿ 0.05).

## 7 Generate Antibody Variants with Strong Affinity to H1N1 Influenza Virus (details)

### Input data

The potent neutralizing antibody F045-092 targets the hemagglutinin (HA) head domain of influenza H3N2 subtypes, as shown in crystallographic structures (PDB IDs: 4O58 and 4O5I). However, no experimentally resolved structure exists for F045-092 in complex with H1N1 subtypes. As a case study to demonstrate the utility of benchmarking models for in-silico affinity maturation, we employed AlphaFold 3 to predict the antibody-antigen complex, aiming to recapitulate native binding

interactions. Specifically, we provided the heavy and light chain sequences of F045-092 along with the HA protein sequence from the 2009 pandemic H1N1 strain (A/California/07/2009). The resulting complex structure achieved a mean pLDDT of 83.47, an inter-chain predicted TM-score (iPTM) of 0.39, and a predicted TM-score (PTM) of 0.49.

DESIGN AND SAMPLING OF CDR-H3 LOOP

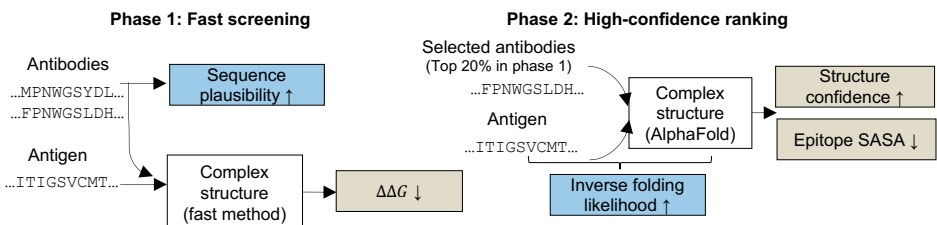

Figure S1: Evaluation of new antibody design. Left: assess sequence plausibility and binding energy change. Right: high-confidence ranking based on AF3-predicted complex structures, evaluating complex structure confidence, epitope SASA, and inverse folding likelihood.

We sampled antibody variants with mutation in the CDR-H3 loop. Each model was provided with the AlphaFold 3-predicted Ab-Ag complex structure described above as input. Specific configuration details for each sampling approach are outlined below:

- **MEAN Sampling** `MEAN` is an antibody sequence-structure co-design model based on an E(3)-equivariant graph neural network (GNN). Trained to jointly predict masked sequence and structure information from antibody data, MEAN can generate mutations with high likelihood in specified regions, leveraging its equivariant architecture to maintain structural consistency. To generate mutations, we perform alanine scanning across all residues in the CDR-H3 region to compute the masking probability for each residue, and then pre-specify the masked regions by sampling from these probabilities. Following this, we generate both new structures and sequences through a multi-round generation process, where the entire structure and sequence are generated in one shot rather than in an autoregressive manner, but modified across multiple rounds according to the strategy proposed by the authors.

- **DiffAb Sampling** `DiffAb` uses a diffusion-based generative process to explore the mutational landscape of antibody sequences. Later timesteps in the diffusion process enable broader exploration of sequence space. To capture mutations at various levels of perturbation, we sampled CDR-H3 sequences at multiple timepoints $(t = 1, 2, 4, 8)$, which allows for both conservative and aggressive mutational strategies. For consistency, sampled variants with more than 5 mutations were excluded. As `diffAb` sometimes outputs wild-type sequences with different loop formations, we repeated the sampling process with different seed values up to 15 to diversify the sequence selection.

- **ESM-IF Sampling** `ESM-IF` provides log-probabilities of amino acid residues conditioned on a given wild-type structure. For sampling purposes, we performed in-silico deep mutagenesis scanning and obtained log-likelihood scores for each single position across 19 other amino acids. For single-point mutations with higher scores than the wild-type, we then performed a combinatorial selection, randomly choosing combinations of 2, 3, 4, or 5 mutations to form multi-mutant sequences, following the procedure described in Shanker et al's work Shanker et al. (2024). For each combinatorial selection, we repeated it 5 times to enable diverse selection.

- **SaProt sampling** `SaProt` is a protein language model that utilizes both sequence tokens and structure tokens. To achieve this, we retrieve the structure tokens using Fold-Seek Van Kempen et al. (2024). As with the mutation strategy used for MEAN, we pre-specify the masked region for mutations based on alanine scanning results, and the total number of mutations ranges from one to five. We then mask only the sequence tokens to generate a new antibody sequence while preserving the original structure.

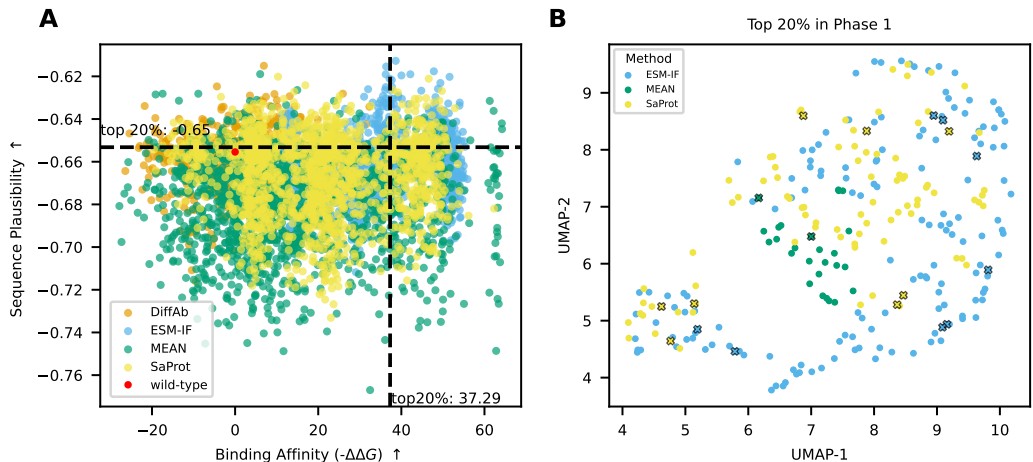

Figure S2: **A.** Binding potential and sequence plausibility in Phase 1. Binding potential was presented as negative binding energy changes $-\Delta\Delta G$, and $\Delta\Delta G$ is defined as $\Delta G_{variant} - \Delta G_{wild}$, where $\Delta G_{wild} = 66.34$. The top 20% of variants, defined by plausibility and binding affinity, are shown in the upper right corner. **B.** Top 20 % variants' CDR-H3 sequence diversity by a UMAP plot of sequence embeddings. 18 non-dominated Pareto-optimal variants, which are marked with a $\times$ symbol, were identified by considering all five metrics on binding potential and sequence plausibility.

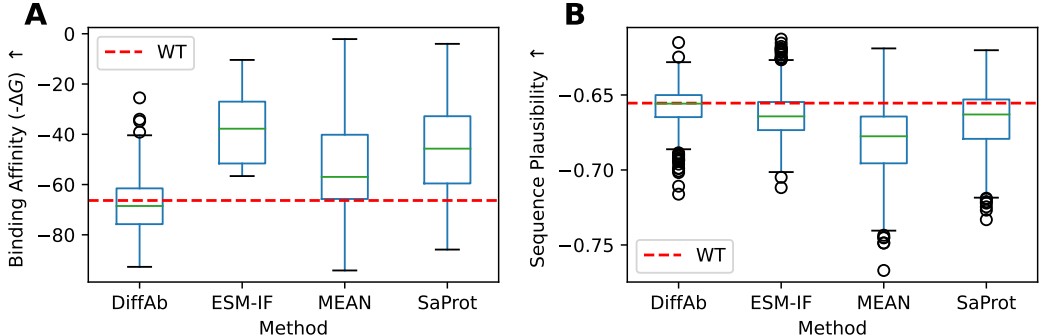

Figure S3: **A.** Boxplots of the binding energy ($-\Delta G$). The wild type is -66.340. **B.** Boxplots of biological plausibility of model-predicted antibody sequences. The wild type is -0.655

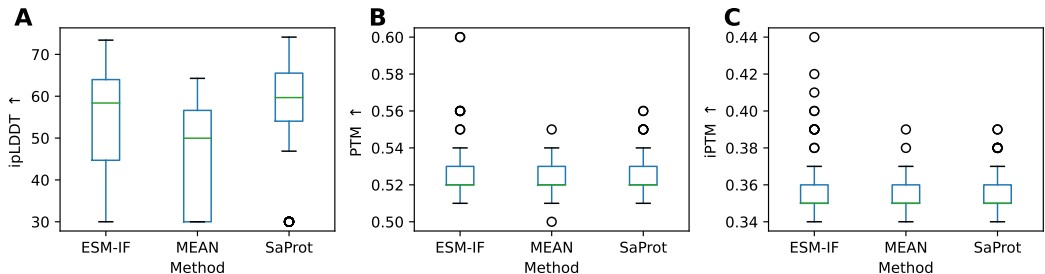

Figure S4: **A.** Comparing the accuracies of side-chain orientations within the binding interface. **B.** Comparing the global structural confidence of the entire complex structure. **C.** Comparing the global structural confidence of interfacial residues.

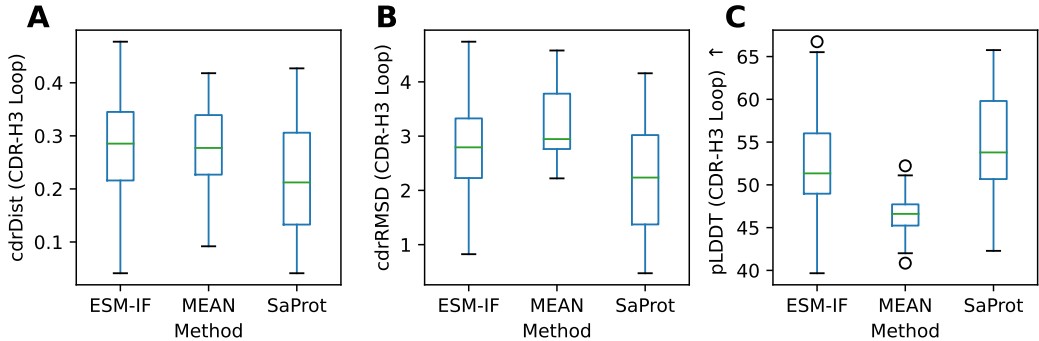

Figure S5: **A.** Comparing sequence similarity. **B.** Comparing loop conformation similarity. **C.** Comparing against the accuracies of side-chain orientations of the CDRH3 residues.

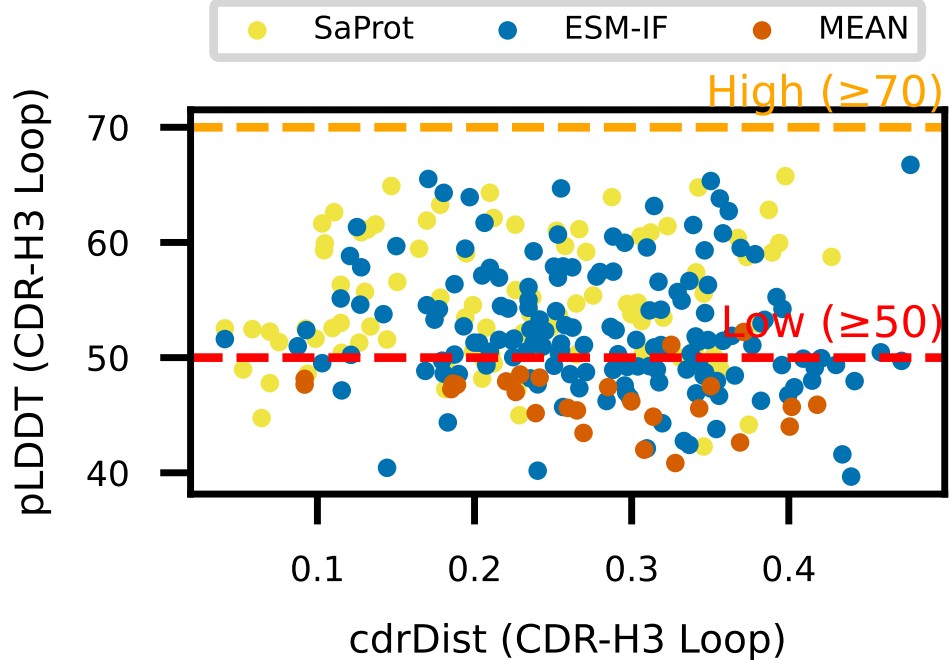

Figure S6: A scatter plot of pLDDT and sequence divergence indicated by cdrDist. Phase 1-screened variants were used for this analysis.

## 8 LIMITATIONS

This study has several limitations. First, the absence of experimental neutralization readouts (e.g., $IC_{50}$) restricts the ability of generative models to design therapeutic antibodies with tighter binding and stronger potency. Therefore, future studies should incorporate complementary functional assay data, providing more biologically relevant training signals for generative models and enhancing the reliability of their predictions. Second, our reliance on purely computational metrics to estimate N-fold binding may not fully capture real-world binding behavior. In-depth structural analyses and direct experimental validation remain essential for confirming actual receptor-antibody interactions and validating computational predictions. Third, while our benchmark includes diverse antigens and

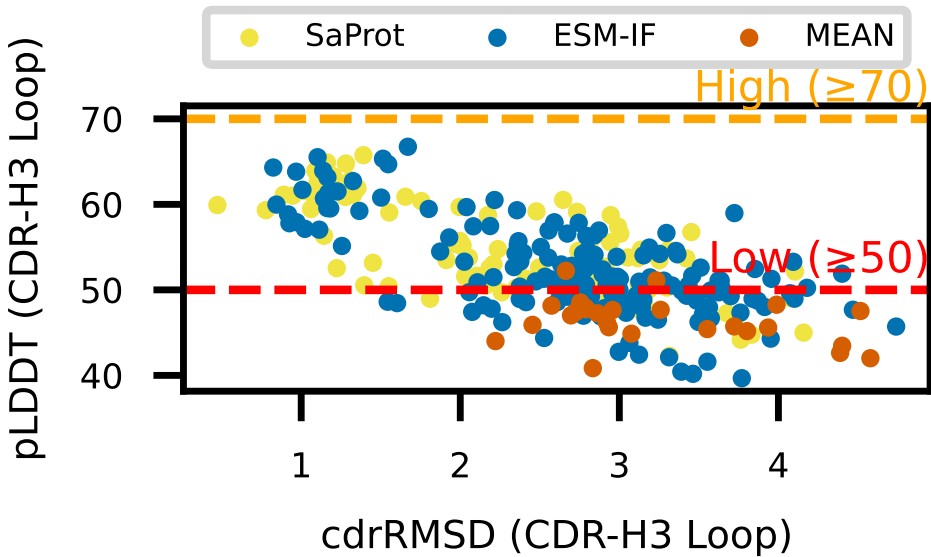

Figure S7: Relationship between structure confidence (pLDDT) and structural deviation from wild type (cdrRMSD) of CDR-H3 generated by models.

nearly 150K antibody variants, some datasets (e.g., 1mhp, 1n8z) are limited in size or derived from narrow mutational libraries, potentially impacting the statistical power of per-dataset evaluations.

To address these limitations, future work will focus on expanding experimental datasets, incorporating functional readouts, and increasing the diversity and scale of benchmark tasks. These improvements will enable more accurate and biologically grounded evaluation of generative antibody design models.

