# OpenReview forum: "AbBiBench: A Benchmark for Antibody Binding Affinity Maturation and Design"
_ICLR.cc/2026/Conference — Submitted to ICLR 2026_

### Official Review · Reviewer_jRK3 · 2025-10-29

**Soundness:** 2
**Presentation:** 2
**Contribution:** 3
**Rating:** 2
**Confidence:** 3

**Summary:**

The manuscript introduces AbBiBench (Antibody Binding Benchmarking), a comprehensive benchmarking framework specifically tailored for the tasks of antibody binding affinity maturation and design. A core principle of this framework is the treatment of the antibody–antigen (Ab–Ag) complex as the fundamental functional unit for analysis. The authors curate an extensive dataset comprising over 186,580 experimental measurements of antibody mutants spanning 13 antibodies and 9 antigens. The paper systematically evaluates 15 distinct classes of protein models—including masked language models, autoregressive models, inverse folding models, diffusion-based generative models, and geometric graph models.

**Strengths:**

1.  **Creation of a Comprehensive Dataset**: The assembly and curation of the AbBiBench dataset constitute a significant and valuable contribution to the field of computational antibody engineering. By integrating a large volume of experimental binding affinity measurements across a diverse set of clinically relevant antibody–antigen complexes, the authors provide a crucial resource necessary for robustly training and rigorously evaluating next-generation generative and predictive models for affinity maturation.

**Weaknesses:**

1. **Omission of Relevant Baselines:** The empirical comparison lacks the inclusion of recent, highly relevant structural prediction models, most notably those in the AlphaFold3-series. Given that several recent studies have demonstrated the utility of confidence metrics derived from these advanced structure prediction pipelines (e.g., ipTMs) for estimating the impact of mutations on protein-protein interaction stability and binding affinity [1, 2].
2. **Limited Predictive Significance of Observed Correlations**: The reported correlation coefficients ($\rho$) in Section 4.1 are consistently low, often falling between 0.1 and 0.2. While the authors attempt to differentiate performance within this narrow band, the statistical and predictive difference between, for example, $\rho=0.2$ and $\rho=0.1$ is arguably too slight to be considered meaningful or indicative of strong predictive power for practical affinity maturation.

**Questions:**

1. The citation style throughout the manuscript appears inconsistent with standard academic conventions. Could the authors please verify and uniformly revise the citation format to adhere to a recognized academic standard?

2. In Section 3.3.2, could the authors elaborate on the practical steps taken to determine $\Delta G$ without structure prediction?

3. To enhance the conviction and completeness of the benchmark results, would the authors consider incorporating the performance metrics (e.g., correlation with affinity) derived from AlphaFold2 and AlphaFold3?



[1] Wee, J. and Wei, G.W., 2024. Evaluation of AlphaFold 3’s protein–protein complexes for predicting binding free energy changes upon mutation. Journal of Chemical Information and Modeling, 64(16), pp.6676-6683.

[2] Lu, Wei, Jixian Zhang, Jiahua Rao, Zhongyue Zhang, and Shuangjia Zheng. "AlphaFold3, a secret sauce for predicting mutational effects on protein-protein interactions." bioRxiv (2024): 2024-05.

---

> ### Author Response · Authors · 2025-12-01
>
> We thank the reviewer for the careful assessment and address the concern as follows.
>
> **W1. Omission of Relevant Baselines (AlphaFold3-series / confidence metrics)**
>
> Thank you for highlighting this important point. We agree that modern structure prediction pipelines (e.g., AlphaFold2/AlphaFold3-style models) and their confidence signals (such as ipTM) are highly relevant baselines for estimating affinity in antibody–antigen interactions. In response, we have begun adding two representative, state-of-the-art structure-prediction baselines—AlphaFold3 (using the wild-type MSA for all mutants to reduce computational cost) and Boltz-2—to complement the current evaluation set.
>
> Because structure prediction across the full AbBiBench scale is computationally expensive, we could only complete partial-dataset experiments before the rebuttal deadline. Nevertheless, these preliminary results already indicate that confidence signals are not reliable predictors of mutant–antigen binding affinity. Full evaluations will be included in the updated manuscript and leaderboard.
>
> Table 2. Preliminary correlations between ipTM and experimental affinity:
> | ID         | AF3 (fixed MSA) | Boltz-2 |
> |------------|------------------|---------|
> | 1mhp       | -0.5323          | -0.2160 |
> | 1mlc       | -0.1736          | 0.0311  |
> | 1n8z       | -0.1571          | -0.0632 |
> | 5a12_ang2  | -0.1089          | -0.0152 |
> | aayl49_ML  | 0.0048           | 0.0198  |
> | aayl49     | 0.2335           | 0.0169  |
> | aayl51     | 0.0654           | 0.0475  |
> | 2fjg       | 0.0914           | 0.0860  |
> | **Average**| **-0.0721**      | **-0.0116** |
>
>
>
>
>
> **W2. Limited Predictive Significance of Low Correlations (ρ ≈ 0.1–0.2)**
>
> We agree that the absolute correlation magnitudes reported in Section 4.1 are modest. We would like to clarify the intended interpretation: AbBiBench aims to provide a comparative benchmark under realistic experimental conditions, not to claim that any single scalar score can consistently deliver strong absolute affinity prediction across all antibody–antigen contexts.
>
> In addition, modest correlations are expected in large, multi-source mutational affinity collections. The observed performance highlights that affinity maturation and optimization remain challenging, motivating the need for standardized, complex-aware benchmarks such as AbBiBench.
>
> **Q1. Citation style inconsistency**
>
> Thank you for noticing this. We will revise all citations and bibliography entries to follow a single venue-consistent academic style.
>
> **Q2. Section 3.3.2: practical steps to determine ΔG without structure prediction**
>
> Thank you for the question. We have clarified the two-phase filter used in the case study. Binding-energy (ΔG) calculations rely on structural input. In Phase 1, we used model-predicted structures from each structure-based model to estimate ΔG. In Phase 2, AlphaFold 3 was used to generate high-confidence Ab–Ag complex structures for refined screening. The revised text now accurately reflects this workflow.
>
> Original text:
> “In Phase 1 we evaluated all 1,500 variants per model using sequence plausibility (AntiBERTy likelihood) and ∆G– metrics that do not require structure prediction…”
>
> Revised text:
> “In Phase 1 we evaluated all 1,500 variants per model using sequence plausibility (AntiBERTy likelihood) and ΔG derived from model-predicted binding poses, and selected the top 20% variants. In Phase 2, we generated full Ab–Ag complex structures with AlphaFold 3 for this subset and computed pLDDT, epitope SASA, and inverse-folding likelihood.”
>
> **Q3. Incorporate AF2/AF3 performance metrics (correlation with affinity)**
>
> Please refer to our response to W1.

---

### Official Review · Reviewer_gK9s · 2025-10-29

**Soundness:** 2
**Presentation:** 3
**Contribution:** 2
**Rating:** 2
**Confidence:** 5

**Summary:**

This paper introduces AbBiBench, benchmark dataset and framework for antibody affinity maturation and design with focus on the regions of the data landscape that have accompanying structural information available. Within the designed framework authors compare several publicly available methods and also pursue an experimental validation on clinically relevant target.

**Strengths:**

The paper is clear and easy to follow with only some minor unclear parts (especially section 2.1). Authors idea on the benchmarking framework and the benchmark itself is valuable and could be useful in the field. Paper also performs experimental validation which is a big strength in my opinion.

**Weaknesses:**

The main weakness of the paper, in my opinion, is related to the scarcity and lack of novelty in terms of the aggregated datasets that are subsequently used in benchmarks. Authors present 15 individual datasets (coming from around 8 individual research articles) featuring roughly 200k measurements. Although this number may seem impressive, virtually each of these datasets was already used as a benchmark in several ML-related publications and majority of them are available publicly in easy to parse formats. Showing-off number of measurements can also be misleading since data comes from a variety of assays (high- and low-throughput and high-throughput results may not necceserily correspond to more accurates ones). Some publicly available datasets, recognized in the field, are also not included for unclear reason - e.g. IgDesign 2025 dataset. While I endorse the idea of the paper, I believe the execution lacks the detail and therefore prevents me from scoring this submission higher. I would welcome and score much higher the submission that would add much more value through e.g. manual (or LLM?) inspection of a large number of manuscripts and extracting the valuable data hiding there for many targets, similar to the effort in e.g. Skempi database.

**Questions:**

- Is there any particular reason that some datasets with structural support were omitted by authors, e.g. Dreyer et al, 2024, Shanehsazzadeh et al, 2024?

---

> ### Author Response · Authors · 2025-12-01
>
> We thank the reviewer for the careful assessment and address the concern as follows.
>
> **W1. Novelty/misleading measurement counts**
>
> Table 1. Overlap vs. AbBiBench-unique datasets (by benchmark)
> | Prior work | Datasets overlapping with **AbBiBench** | Datasets unique to **AbBiBench** vs. this work |
> |------------|------------------------------------------|--------------------------------------------------|
> | **FLAb** | VEGF–G6.31 (2FJG); Lysozyme–D44.1 (1MLC); HER2–trastuzumab (1N8Z) | Influenza HA: 4FQI_H1 / 4FQI_H3 / 3GBN_H1 / 3GBN_H9; SARS-CoV-2 HR2: AAYL49/50/51/52, AAYL49_ML; Integrin αIIbβ3: 1MHP; Ang2: 5A12_Ang2 |
> | **IgDesign** | HER2–trastuzumab (1N8Z) | VEGF (2FJG); Lysozyme (1MLC); Influenza HA (4FQI/3GBN series); SARS-CoV-2 HR2 (AAYL49/50/51/52 + AAYL49_ML); Integrin (1MHP); Ang2 (5A12_Ang2) |
> | **ProteinGym** | None (ProteinGym is not an antibody–antigen affinity benchmark) | All AbBiBench datasets are unique vs. ProteinGym |
> | **AbRank** | AAYL49/50/51/52; HER2–trastuzumab (1N8Z) | Influenza HA (4FQI/3GBN series); VEGF (2FJG); Lysozyme (1MLC); Integrin (1MHP); Ang2 (5A12_Ang2); AAYL49_ML |
>
> We respectfully disagree that AbBiBench’s contribution is primarily “showing off” already-benchmarked data. While individual assays may appeared in prior works,  As shown in Table 1, the datasets that AbBiBench provides but none of the other benchmarks include are:
> Influenza HA systems (4FQI_H1, 4FQI_H3, 3GBN_H1, 3GBN_H9),
> Integrin αIIbβ3 (1MHP),
> Ang2 (5A12_Ang2),
> and AAYL49_ML.
> These datasets were newly identified and curated by us through a targeted literature search.
>
> **W2. Omission of “publicly recognized datasets”, e.g., IgDesign 2025**
>
> Thank you—this is helpful. As an extensible benchmark, we will include the IgDesign dataset in the next version and cite that work. Besides, for the dataset you mentioned from Shanehsazzadeh et al. (2024), we have already included it as the HER2/trastuzumab affinity dataset (ID 1n8z).
>
> **W3. Request for added “value” via manual/LLM extraction like SKEMPI-style curation**
>
> We agree this is a high-impact direction. The current submission focuses on a clean, reproducible, complex-based benchmark with standardized and public evaluation code, plus an end-to-end generative case study with wet-lab ELISA validation.
>
> To address the reviewer’s request, we will expand the Limitations/Future Work to explicitly propose LLM-assisted literature mining to extract additional affinity landscapes across more targets, SKEMPI-style, while maintaining auditability and provenance.
>
> **Q1. Why omit structural datasets like Dreyer et al., 2024; Shanehsazzadeh et al., 2024?**
>
> Thank you for this helpful suggestion. Please refer to our response to W2.

---

### Official Review · Reviewer_826a · 2025-11-01

**Soundness:** 2
**Presentation:** 2
**Contribution:** 1
**Rating:** 2
**Confidence:** 4

**Summary:**

This paper introduces AbBiBench, a large-scale benchmarking framework for antibody binding affinity maturation and design. Unlike prior metrics that evaluate antibodies in isolation, AbBiBench evaluates antibody–antigen complexes together. It comprise of experimental data from over 186,000 mutants across 13 antibodies and 9 antigens and systematically compares 15 protein models. Results show that structure-conditioned inverse folding models outperform others in predicting and generating high-affinity variants.

**Strengths:**

1. The benchmark consists of a wide range of antibodies, antigens, and model architectures, which allows comprehensive and biologically informed evaluation for antibody design models.
2. The study includes in vitro validation, providing strong experimental evidence that supports the findings of the computational benchmark.

**Weaknesses:**

1. While AbBiBench provides a biologically meaningful benchmarking pipeline, it primarily integrates existing protein and antibody machine learning models without introducing novel machine learning methodologies.
2. The antibody generation by sampling from the models focuses on a single antigen influenza H1N1. It limits the generalizability of the generation results across diverse antigen targets.

**Questions:**

1. What is the correlation between affinity fold change and actual binding affinity? Why is it better than other latest binding affinity prediction models?
2. The Phase 2 for identifying final candidates in 3.3.2 relies on AlphaFold 3 for full complex structure generation, which is a computationally intensive step. Could the author provide details and discussion on the computational time/efficiency of the benchmark?

---

> ### Author Response · Authors · 2025-12-01
>
> Thank you for the reviewer’s thoughtful comment. We respectfully clarify the following points.
>
> **W1. No novel machine learning methodology; mainly integrates existing models.**
>
> We respectfully disagree.
>
> *First*, the ICLR 2026 Call for Papers explicitly includes “datasets and benchmarks” as a core submission area, confirming that benchmark contributions are fully valid and encouraged.
>
> *Second*, ICLR routinely accepts benchmark-focused papers that do not introduce new models—for example, VTDexManip and GameArena (ICLR 2025), whose contributions lie in data curation and systematic evaluation rather than model innovation.
>
> *Third*, the ICLR Reviewer Guide evaluates submissions by clarity, rigor, correctness, motivation, and value to the community. It does not require introducing a new SOTA model; rather, it emphasizes providing “new, relevant, impactful knowledge,” which directly covers benchmark papers.
>
> *Novelty of AbBiBench*
>
> First complex-based, large-scale Ab–Ag benchmark.
> Prior benchmarks mostly evaluate antibodies in isolation. AbBiBench standardizes 14 datasets (≈184k variants) at the complex level with paired sequences, structures, and affinity measurements, requiring substantial cross-study curation.
>
> Unified, biology-informed evaluation.
> We provide a standardized protocol mapping 15 diverse models (PLMs, autoregressive, inverse folding, diffusion, graph, physics-based) to the same Ab–Ag complex input and zero-shot affinity prediction interface.
>
> End-to-end design pipeline with wet-lab validation.
> Beyond prediction benchmarking, we operationalize a full design workflow and experimentally validate that top-ranked models can drive successful affinity-improving designs—an aspect missing from existing antibody ML benchmarks.
>
> Clear scientific findings.
> Structure-conditioned inverse-folding models consistently achieve the best affinity correlations and generation performance, offering actionable insights for future antibody-model development.
>
> In sum, AbBiBench is not “just running existing models”; it establishes a complex-based experimental standard—curated data, unified evaluation, and validated design workflows—that the community can directly build upon.
>
> **W2. Generation focuses on a single antigen (H1N1), limiting generalizability**
>
> Our main claims—including affinity-correlation analyses and model ranking—are based on zero-shot prediction across 11 antibodies and 9 antigens (Table 1), already covering diverse antigen contexts.
>
> For generation, wet-lab validation of designed antibodies is extremely rare in ICLR. Rather than limiting generalizability, it provides stronger evidence: top-ranked models identified by AbBiBench produce mutations that bind in vitro. Each antigen requires full cloning–expression–purification–assay cycles, so even one validated system has high biological value.
>
> **Q1. Correlation between affinity fold change and actual affinity? Why better than latest models?**
>
> Definition.
> Affinity fold change derives from experimental pKd:
>
> $\text{fold change} = 10^{(pK_{d,\text{mut}} - pK_{d,\text{WT}})}$
>
>
> Thus, it is a monotonic transform of Kd and corresponds exactly to binding strength.
>
> What the benchmark evaluates.
> We compute Spearman correlation between model scores and experimental affinity for each dataset, and assess enrichment for ≥5-fold improvements (precision@10). Inverse-folding models consistently show the highest average correlations and enrichment across 14 datasets.
>
> Regarding “better than latest affinity models.”
> We do not introduce a new model; we provide the first large-scale, complex-based benchmark enabling fair comparison. The conclusion follows directly from standardized evaluation.
>
> **Q2. AF3 in Phase 2 is computationally intensive; please discuss runtime.**
>
> AF3 is used only in Phase 2 on a down-selected subset. We first generate 1,500 variants per model and filter using AntiBERTy likelihood and coarse biophysics metrics. Only the top 20% proceed to AF3 evaluation for structure prediction and metrics (pLDDT, epitope SASA, inverse-folding likelihood).
>
> Predicting one complex with AF3 takes 3,891 seconds on one H100 GPU.
>
> Core benchmark remains lightweight.
> All zero-shot inference over ≈184k mutants for each model runs on a single H100 GPU with no training or fine-tuning. Thus, the benchmark is efficient and reproducible even without large compute resources.

---

### Official Review · Reviewer_F8BK · 2025-11-01

**Soundness:** 3
**Presentation:** 3
**Contribution:** 4
**Rating:** 6
**Confidence:** 4

**Summary:**

The authors introduce AbBiBench, a benchmarking framework for antibodies. It evaluates models by scoring the complete antibody-antigen complex rather than the antibody in isolation, arguing this is a more biologically sound approach. Using a curated dataset of over 186,580 experimental affinity measurements, this paper benchmarks 15 protein models and finds that structure-conditioned inverse folding models (like ESM-IF) generally outperform other architectures. This conclusion is supported by an in vitro case study where designed antibodies successfully gained a new binding function.

**Strengths:**

- Assembling, curating, and standardizing over 186,580 experimental measurements is a very appreciated contribution to the field. Making this dataset public will accelerate future model development.
- The authors performed in vitro ELISA assays on 21 designed variants and showed a clear gain-of-function (H1N1 binding) that the wild-type antibody lacked. This validates that the benchmark's top models can be used in a practical, successful design campaign.

**Weaknesses:**

- The benchmark, and its in vitro validation, focuses on binding affinity (Kd or ELISA OD signals). In a therapeutic context, the ultimate goal is function (e.g., neutralization, measured by IC50). But affinity is a common proxy used in most computational studies.
- While the benchmark's focus on binding affinity is important, it doesn't capture the full picture. Antibody design is a multi-parameter optimization problem, and the authors acknowledge their work would be more beneficial if it included other key data like stability, immunogenicity, and functional assays.
- The generative case study (against H1N1) is based on an AlphaFold3-predicted structure, as no experimental one exists. The paper's supplement reveals this predicted complex has an iPTM score of 0.39, which indicates very low confidence in the predicted interface.
- Two anti-influenza antibodies (CR9114 and CR6261) account for most of the dataset. This means the "Average" performance (Figure 3) is overwhelmingly dominated by how well models perform on influenza, not on general Ab-Ag interactions.

**Questions:**

- It would be great to incorporate data beyond binding affinity, e.g. from Flab, in the benchmark to allow for multi-parameter optimization and assessment
- Some class balancing in assessing overall performance across the different, very unbalanced datasets, would be useful, as well as some discussion of epitope diversity in the dataset.

---

> ### Author Response · Authors · 2025-12-01
>
> We thank the reviewer for the careful assessment and address the concern as follows.
>
> **W1. Affinity vs. therapeutic function**
>
> We agree that therapeutic efficacy is ultimately determined by functional readouts (e.g., neutralization IC50), and binding affinity is only a proxy. AbBiBench focuses on affinity maturation because affinity measurements are the most available, informative, and standardized signals across studies, enabling a benchmark of this scale. Affinity also offers a unique computational challenge: evaluating it requires modeling the antibody–antigen complex, which tests a model’s ability to capture interface geometry, long-range interactions, and sequence–structure coupling. We acknowledge this limitation and plan to include functional assay data such as IC50 in future versions.
>
> **W2. Antibody design is multi-parameter**
>
> We agree. Antibody design spans multiple objectives—affinity, specificity, stability, solubility, developability, immunogenicity, and ultimately in vivo function. Yet affinity is the primary organizing axis: if affinity is insufficient, improvements in other properties cannot rescue activity. It is also the only property consistently available at scale across public datasets. Because affinity evaluation requires modeling the full Ab–Ag complex, it is a stringent target for protein models and motivates the attention given to complex-level assessment in AbBiBench.
>
> We also position this as future work: expanding AbBiBench to include stability, immunogenicity, developability, and functional assays. Relatedly, FLAb is a valuable complementary resource for non-affinity properties, but it does not incorporate antigen context for affinity—the specific gap AbBiBench addresses.
>
> **W3. Case study uses an AlphaFold3-predicted complex with low iPTM**
>
> We appreciate this concern. We report the AF3 metrics transparently (mean pLDDT 83.47; iPTM 0.39; PTM 0.49).
>
> Two clarifications:
> • The limited iPTM motivates choosing this H1N1 system for optimization; if interface confidence were already high, affinity design would be less meaningful.
> • We do not treat the AF3 structure as ground truth. Instead, we use a multi-metric, two-phase scheme (structure confidence + epitope SASA + inverse-folding likelihood) to avoid reliance on any single uncertain signal.
>
> Most importantly, functional gain is supported by wet-lab ELISA assays across all 21 designed mutants, which validates improved binding independently of AF3 confidence.
>
> **W4. Dataset imbalance: influenza (CR9114/CR6261) dominates overall “Average”**
>
> Although datasets differ in size, AbBiBench uses the mean of per-dataset scores—not size-weighted aggregation—to compute model rankings. This macro-averaging prevents large influenza libraries from dominating results and ensures each dataset contributes equally.
>
> We will continue curating additional antibody–antigen datasets to further balance diversity in future releases.
>
> **Q1. Incorporate more than affinity (e.g., FLAb) for multi-parameter optimization**
>
> We thank the reviewer. As noted above, we agree that incorporating stability, immunogenicity, and functional assays will strengthen AbBiBench. However, affinity remains the biologically foundational property and the only signal consistently available across large public datasets, enabling a rigorous benchmark of this scale. We view AbBiBench as an extensible framework and will integrate additional modalities as datasets mature.
>
> **Q2. Class balancing + epitope diversity discussion**
>
> We appreciate the feedback. As discussed in W4, AbBiBench mitigates size imbalance by macro-averaging per-dataset scores, preventing influenza datasets from dominating results. We also agree that epitope diversity is important, and we will continue curating new antibody–antigen datasets to enhance diversity and balance in future versions.

---

### Meta-Review · Area_Chair_nF9t · 2026-01-06

**Summary:**

The main concern of this paper is that it merely integrates existing protein and antibody machine learning models without introducing novel machine learning methodologies.

**Reviewer Concerns:**

The reviewer concerns were not adequately addressed by the reviewer.

**Reviewer Scores:**

The reviewer would have kept their score even if they participated fully in the discussion.

---

### Decision · Program_Chairs · 2026-01-26

Reject